# Why Not to Use Zero Imputation? Correcting Sparsity Bias in Training Neural Networks

**Joonyoung Yi[1], Juhyuk Lee[1], Kwang Joon Kim[3], Sung Ju Hwang[1,2], Eunho Yang[1,2]**
KAIST[1], AITRICS[2], Yonsei University College of Medicine[3], South Korea
`{joonyoung.yi, sehkmg, sjhwang82, eunhoy} @kaist.ac.kr,`
`preppie@yuhs.ac`

## Abstract

Handling missing data is one of the most fundamental problems in machine learning. Among many approaches, the simplest and most intuitive way is zero imputation, which treats the value of a missing entry simply as zero. However, many studies have experimentally confirmed that zero imputation results in suboptimal performances in training neural networks. Yet, none of the existing work has explained what brings such performance degradations. In this paper, we introduce the *variable sparsity problem (VSP)*, which describes a phenomenon where the output of a predictive model largely varies with respect to the rate of missingness in the given input, and show that it adversarially affects the model performance. We first theoretically analyze this phenomenon and propose a simple yet effective technique to handle missingness, which we refer to as *Sparsity Normalization (SN)*, that directly targets and resolves the VSP. We further experimentally validate SN on diverse benchmark datasets, to show that debiasing the effect of input-level sparsity improves the performance and stabilizes the training of neural networks.

## 1 Introduction

Many real-world datasets often contain data instances whose subset of input features is missing. While various imputing techniques, from imputing using global statistics such as mean, to individually imputing by learning auxiliary models such as GAN, can be applied with their own pros and cons, the most simple and natural way to do this is *zero imputation*, where we simply treat a missing feature as zero. In neural networks, at first glance, zero imputation can be thought of as a reasonable solution since it simply drops missing input nodes by preventing the weights associated with them from being updated. Some what surprisingly, however, many previous studies have reported that this intuitive approach has an adverse effect on model performances (Hazan et al., 2015; Luo et al., 2018; Śmieja et al., 2018), and none of them has investigated the reasons of such performance degradations.

In this work, we find that zero imputation causes the output of a neural network to largely vary with respect to the number of missing entries in the input. We name this phenomenon *Variable Sparsity Problem (VSP)*, which should be avoided in many real-world tasks. Consider a movie recommender system, for instance. It is not desirable that users get different average of predicted ratings just because they have rated different number of movies (regardless of their actual rating values). One might argue that people with less ratings do not like movies in general and it is natural to give higher predicted values to people with more ratings. This might be partially true for users of *some* sparsity levels, but it is not a common case uniformly applicable for a wider range of sparsity levels. This can be verified in real collaborative filtering datasets as shown in Figure 1 (upper left corner) where users have a similar average rating for test data regardless of the number of known ratings (see also other two examples in Figure 1). However, in standard neural networks with zero imputation, we observe that the model's inference correlates with the number of known entries of the data instance as shown in

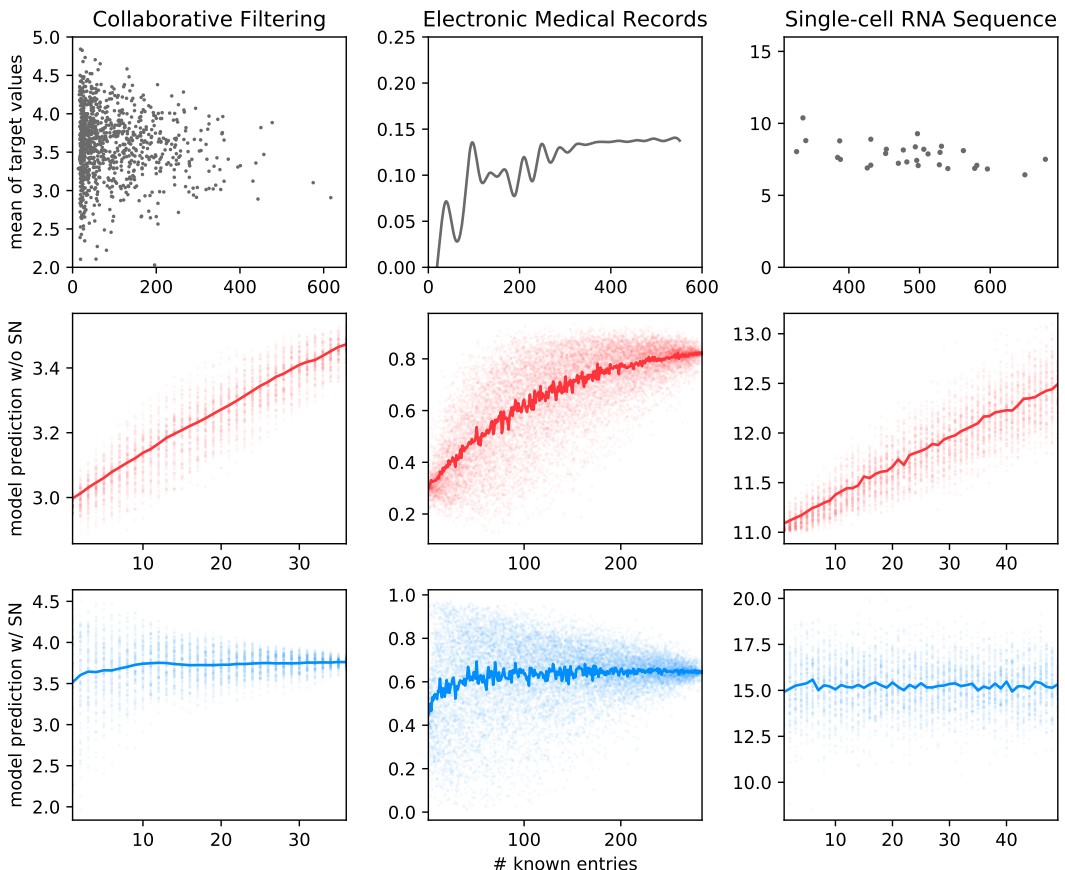

Figure 1: **First column**: Sedhain et al. (2015) on Movielens 100K (collaborative filtering) dataset. **Second column**: LSTM on Physionet 2012 (electronic medical records) dataset. **Third column**: Talwar et al. (2018) on Blakeley (single cell RNA sequence) dataset. **First row**: Mean of target values according to the number of known entries in training set. **Second row**: Predicted values of models with zero imputation according to the number of known entries for a randomly selected test point. Input masks are randomly sampled (to artificially control its sparsity level). For each target sparsity level through x-axis, 50 samples are drawn, scattering the predicted values and plotting the average in solid line. **Third row**: Figures on how plots in second row are corrected by Sparsity Normalization.

the second row of Figure 1[1]. It would be fatal in some safety-critical applications such as a medical domain: a patient's probability of developing disease for example should not be evaluated differently depending on the number of medical tests they received (we do not want our model to predict the probability of death is high just because some patient has been screened a lot!).

In addition, we theoretically analyze the existence of VSP under several circumstances and propose a simple yet effective means to suppress VSP while retaining the intuitive advantages of zero imputation: normalizing with the number of non-zero entries for each data instance. We refer to this regularization as *Sparsity Normalization*, and show that it effectively deals away with the VSP, resulting in significant improvements in both the performance and the stability of training neural networks.

Our contribution in this paper is threefold:

- To best of our knowledge, we are the first in exploring the adverse effect of zero imputation, both theoretically and empirically.

---

[1]Note that, this tendency is very consistent with other test points and is observed throughout the entire learning process (even before the training).

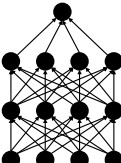 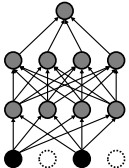 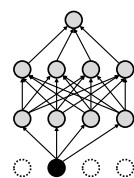 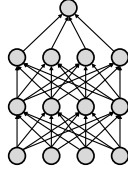 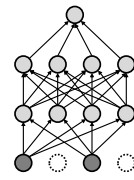 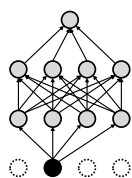

(a) Neural network with variable sparsity levels    (b) with Sparsity Normalization

Figure 2: The change of output values according to the sparsity level of inputs (darker color indicates greater absolute value and dotted circles indicate missing nodes with zero imputation). SN makes the possible output range of a network be more stable with respect to the sparsity level.

- We identify the cause of adverse effect of zero imputation, which we refer to as variable sparsity problem, and formally describe how this problem actually affects training and inference of neural networks (Section 2). We further provide new perspectives using VSP to understand phenomena that have not been clearly explained or that we have misunderstood (Section 4 and 5).

- We present Sparsity Normalization (SN) and theoretically show that SN can solve the VSP under certain conditions (Section 3). We also experimentally reaffirm that simply applying SN can effectively alleviate or solve the VSP yielding significant performance gains (Section 4).

## 2 VARIABLE SPARSITY PROBLEM

We formally define the *Variable Sparsity Problem (VSP)* as follows: a phenomenon in which the expected value of the output layer of a neural network (over the weight and input distributions) depends on the sparsity (the number of zero values) of the input data (Figure 2a). With VSP, the activation values of neural networks could become largely different for exactly the same input instance, depending on the number of zero entries; this makes training more difficult and may mislead the model into incorrect predictions.

While zero imputation is intuitive in the sense that it drops the missing input features, we will show that it causes variable sparsity problem for several example cases. Specifically, we show the VSP under assumptions with increasing generality: **(Case 1)** where activation function is an identity mapping with no bias, **(Case 2)** where activation function is an affine function, and **(Case 3)** where activation function is a non-decreasing convex function such as ReLU (Glorot et al., 2011), leaky ReLU (Maas et al., 2013), ELU (Clevert et al., 2016), or Softplus (Dugas et al., 2001).

Here, we summarize the notation for clarity. For a $L$-layer deep network with non-linearity $\sigma$, we use $W^i \in \mathbb{R}^{n_i \times n_{i-1}}$ to denote the weight matrix of $i$-th layer, $\mathbf{b}^i \in \mathbb{R}^{n_i}$ to denote the bias, $\mathbf{h}^i \in \mathbb{R}^{n_i}$ to denote the activation vector. For simplicity, we use $\mathbf{h}^0 \in \mathbb{R}^{n_0}$ and $\mathbf{h}^L \in \mathbb{R}^{n_L}$ to denote input and output layer, respectively. Then, we have

$$\mathbf{h}^i = \sigma(W^i \mathbf{h}^{i-1} + \mathbf{b}^i), \quad \text{for } i = 1, \cdots, L.$$

Our goal in this section is to observe the change in $\mathbf{h}^L$ as the sparsity of $\mathbf{h}^0$ (input $\mathbf{x}$) changes. To simplify the discussion, we consider the following assumption:

**Assumption 1.** *(i) Every coordinate of input vector, $h_l^0$, is generated by the element-wise multiplication of two random variables $\tilde{h}_l^0$ and $m_l$ where $m_l$ is binary mask indicating missing value and $\tilde{h}_l^0$ is a (possibly unobserved) feature value. Here, missing mask $m_l$ is MCAR (missing completely at random), with no dependency with other mask variables or their values $\tilde{\mathbf{h}}^0$. All $m_l$ follow some identical distribution with mean $\mu_m$. (ii) The elements of matrix $W^i$ are mutually independent and follow the identical distribution with mean $\mu_w^i$. Similarly, $\mathbf{b}^i$ and $\tilde{\mathbf{h}}^0$ consist of i.i.d. coordinates with mean $\mu_b^i$ and $\mu_x$, respectively. (iii) $\mu_w^i$ is not zero uniformly over all $i$.*

(i) assumes the simplest missing mechanism. (ii) is similarly defined in Glorot & Bengio (2010) and He et al. (2015) in studying weight initialization techniques. (iii) may not hold under some initialization strategies, but as the learning progresses, it is very likely to hold.

**(Case 1)**   For simplicity, let us first consider networks without the non-linearity nor the bias term. Theorem 1 shows that the average value of the output layer $E[h_l^L]$ is directly proportional to the expectation of the mask vector $\mu_m$:

**Theorem 1.** *Suppose that activation $\sigma$ is an identity function and that $b_l^i$ is uniformly fixed as zero under Assumption 1. Then, we have $E[h_l^L] = \prod_{i=1}^{L} n_{i-1}\mu_w^i \mu_x \mu_m$.*

**(Case 2)**   When the activation function is affine but now with a possibly nonzero bias, $E[h_l^L]$ is influenced by $\mu_m$ in the following way:

**Theorem 2.** *Suppose that activation $\sigma$ is an affine function under Assumption 1. Suppose further that $f_i(x)$ is defined as $\sigma(n_{i-1}\mu_w^i x + \mu_b^i)$. Then, $E[h_l^L] = f_L \circ \cdots \circ f_1(\mu_x \mu_m)$.*

**(Case 3)**   Finally, when the activation function is non-linear but non-decreasing and convex, we can show that $E[h_l^L]$ is lower-bounded by some quantity involving $\mu_m$:

**Theorem 3.** *Suppose that $\sigma$ is a non-decreasing convex function under Assumption 1. Suppose further that $f_i(x)$ is defined as $\sigma(n_{i-1}\mu_w^i x + \mu_b^i)$ and $\mu_w^i > 0$. Then, $E[h_l^L] \geq f_L \circ \cdots \circ f_1(\mu_x \mu_m)$.*

If the expected value of the output layer (or the lower bound of it) depends on the level of sparsity/missingness as in Theorem 1-3, even similar data instances may have different output values depending on their sparsity levels, which would hinder fair and correct inference of the model. As shown in Figure 1 (second row), the VSP can easily occur even in practical settings of training neural networks where the above conditions do not hold.

## 3   SPARSITY NORMALIZATION

In this section, we propose a simple yet surprisingly effective method to resolve the VSP. We first revisit **(Case 2)** to find a way of making expected output independent of input sparsity level since the linearity in activation simplifies the correction. Recalling the notation of $\mathbf{h}^0 = \tilde{\mathbf{h}}^0 \odot \mathbf{m}$ ($\odot$ represents the element-wise product), we find that simply normalizing via $\mathbf{h}_{SN}^0 = (\tilde{\mathbf{h}}^0 \odot \mathbf{m}) \cdot K_1/\mu_m$ for any fixed constant $K_1$, can debias the dependency on the input sparsity level. We name this simple normalizing technique *Sparsity Normalization* (SN) and describe it in Algorithm 1. Conceptually, this method scales the size of each input value according to its sparsity level so that the change in output

---

**Algorithm 1** Sparsity Normalization (SN)

**Input:** Dataset $\mathcal{D}$, constant $K$.
**Output:** Sparsity Normalized Dataset $\mathcal{D}_{SN}$.

Empty set $\mathcal{S} = \phi$
**for** each $(\mathbf{h}^0, \mathbf{m}) \in \mathcal{D}$ **do**
  $\mathbf{h}_{SN}^0 \leftarrow K \cdot \mathbf{h}^0 / \|\mathbf{m}\|_1$
  $\mathcal{S} \leftarrow \mathcal{S} \cup \{\mathbf{h}_{SN}^0\}$
**end for**
$\mathcal{D}_{SN} \leftarrow \mathcal{S}$

---

size is less sensitive to the sparsity level (Figure 2b). The formal description on correcting sparsity bias by SN is as follows in this particular case:

**Theorem 4.** *(With Sparsity Normalization) Suppose that activation $\sigma$ is an affine function under Assumption 1. Suppose further that $f_i(x) = \sigma(n_{i-1}\mu_w^i x + \mu_b^i)$ and replace the input layer using SN, i.e. $\mathbf{h}_{SN}^0 = (\tilde{\mathbf{h}}^0 \odot \mathbf{m}) \cdot K_1/\mu_m$ for any fixed constant $K_1$. Then, we have $E[h_l^L] = f_L \circ \cdots \circ f_1(\mu_x \cdot K_1)$.*

Unlike in Theorem 2, SN in Theorem 4 makes average activation to be independent of $\mu_m$, which determines the sparsity levels of input. It is not trivial to show the counterpart of **(Case 3)** using SN since $E[\sigma(x)] = \sigma(E[x])$ does not hold in general. However, we show through extensive experiments in Section 4 that SN is practically effective even in more general cases.

Table 1: Debiasing variable sparsity using SN on Movielens datasets. Test RMSE with 95% confidence interval of 5-runs is provided.

| Datasets | | | Movielens 100K | Movielens 1M | Movielens 10M |
|---|---|---|---|---|---|
| AutoRec (Sedhain et al., 2015) | user vector | w/o SN | $0.9346 \pm 0.0007$ | $0.8831 \pm 0.0002$ | $0.8859 \pm 0.0014$ |
| | | w/ SN | $\mathbf{0.9208 \pm 0.0023}$ | $\mathbf{0.8742 \pm 0.0003}$ | $\mathbf{0.8462 \pm 0.0005}$ |
| | item vector | w/o SN | $0.8835 \pm 0.0003$ | $0.8320 \pm 0.0003$ | $0.7807 \pm 0.0017$ |
| | | w/ SN | $\mathbf{0.8809 \pm 0.0011}$ | $\mathbf{0.8294 \pm 0.0004}$ | $\mathbf{0.7706 \pm 0.0023}$ |
| CF-NADE (Zheng et al., 2016) | user vector | w/o SN | $0.9253 \pm 0.0010$ | $\mathbf{0.8530 \pm 0.0006}$ | $0.8113 \pm 0.0058$ |
| | | w/ SN | $\mathbf{0.9231 \pm 0.0012}$ | $\mathbf{0.8525 \pm 0.0006}$ | $\mathbf{0.7854 \pm 0.0006}$ |
| | item vector | w/o SN | $0.8982 \pm 0.0005$ | $0.8405 \pm 0.0007$ | N/A |
| | | w/ SN | $\mathbf{0.8900 \pm 0.0018}$ | $\mathbf{0.8366 \pm 0.0008}$ | N/A |
| CF-UIcA (Du et al., 2018) | | w/o SN | $0.8945 \pm 0.0024$ | $0.8223 \pm 0.0016$ | N/A |
| | | w/SN | $\mathbf{0.8793 \pm 0.0017}$ | $\mathbf{0.8178 \pm 0.0007}$ | N/A |

Table 2: Comparison of AutoRec with SN and CF-UIcA with SN against state-of-the-art collaborative filtering methods for each datasets. Bold font indicates neural network based models. The results marked † are taken from Zhang et al. (2017) and all other baselines are taken from original papers. The result of CF-UIcA with SN on Movielens 10M is not provided because the authors of CF-UIcA did not provide the results for it due to the complexity of the model.

| Models | Movielens 100K | Models | Movielens 1M | Models | Movielens 10M |
|---|---|---|---|---|---|
| Koren (2008) | $0.913^{\dagger}$ | Fu et al. (2018) | 0.836 | **Sedhain et al. (2015)** | 0.782 |
| Zhuang et al. (2017) | $0.9114 \pm 0.0093$ | Lee et al. (2016) | 0.8333 | **Berg et al. (2018)** | 0.777 |
| Koren et al. (2009) | $0.911^{\dagger}$ | Yi et al. (2019) | 0.8321 | Chen et al. (2016) | $0.7712 \pm 0.0002$ |
| **Dziugaite & Roy (2015)** | 0.903 | **Berg et al. (2018)** | 0.832 | **Zheng et al. (2016)** | 0.771 |
| Zhang et al. (2017) | 0.901 | **Sedhain et al. (2015)** | 0.831 | **AutoRec w/ SN (ours.)** | $0.7690 \pm 0.0023$ |
| Yi et al. (2019) | 0.8889 | **Zheng et al. (2016)** | 0.829 | Li et al. (2016) | $0.7682 \pm 0.0003$ |
| Lee et al. (2016) | $0.8881 \pm 0.0017$ | **AutoRec w/ SN (ours.)** | $0.8260 \pm 0.0023$ | Chen et al. (2017) | $0.7672 \pm 0.0001$ |
| **AutoRec w/ SN (ours.)** | $0.8816 \pm 0.0087$ | **Du et al. (2018)** | 0.823 | Fu et al. (2018) | 0.766 |
| **CF-UIcA w/ SN (ours.)** | $0.8779 \pm 0.0159$ | **CF-UIcA w/ SN (ours.)** | $0.8215 \pm 0.0037$ | Li et al. (2017) | $0.7634 \pm 0.0002$ |

While Theorem 4 assumes that $\mu_m$ is known and fixed across all data instances, we relax this assumption in practice and consider varying $\mu_m$ across data instances. Specifically, by a maximum likelihood principle, we can estimate $\mu_m$ for each instance by $\left\| \mathbf{h}^0 \right\|_0 / n_0 = \|\mathbf{m}\|_1 / n_0$. Thus, we have $\mathbf{h}_{SN}^0 = K \cdot \mathbf{h}^0 / \|\mathbf{m}\|_1$ where $K = n_0 \cdot K_1$ (see Algorithm 1)[2]. In practice, we recommend using $K$ as the average of $\|\mathbf{m}\|_1$ over all instances in the training set. We could encounter the dying ReLU phenomenon (He et al., 2015) if $K$ is too small (e.g., $K = 1$). Since the hyper-parameter $K$ can bring in a regularization effect via controlling the magnitude of gradient (Salimans & Kingma, 2016), we define $K = E_{(\mathbf{h}^0, \mathbf{m}) \in \mathcal{D}}[\|\mathbf{m}\|_1]$ so that the average scales remain constant before and after the normalization, minimizing such side effects caused by SN.

## 4 Experiments

In this section, we empirically show that VSP occurs in various machine learning tasks and it can be alleviated by SN. In addition, we also show that resolving VSP leads to improved model performance on diverse scenarios.

### 4.1 Collaborative Filtering (Recommendation) Datasets

We identify VSP and the effect of SN on several popular benchmark datasets for collaborative filtering with extremely high missing rates. We train an AutoRec (Sedhain et al., 2015) using user vector on Movielens (Harper & Konstan, 2016) 100K dataset for validating VSP and SN. Going back to the first column of Figure 1, the prediction with SN is almost constant regardless of $\|\mathbf{m}\|_1$. Another perceived phenomenon in Figure 1 is that the higher the $\|\mathbf{m}\|_1$, the smaller the variation in the prediction of

---

[2]When $\|\mathbf{m}\|_1$ is 0, calculation is impossible. Hence, in this case, the $\|\mathbf{m}\|_1$ is assumed to be 1.

Table 3: Debiasing variable sparsity using SN and comparisons against other missing handling techniques on five disease identification tasks of NHIS dataset. Test AUROC with 95% confidence interval of 5-runs is provided.

| Dataset | Cardiovascular | Fatty Liver | Hypertension | Heart Failure | Diabetes |
|---|---|---|---|---|---|
| Zero Imputation w/o SN | $0.7057 \pm 0.0027$ | $0.6750 \pm 0.0050$ | $0.7977 \pm 0.0027$ | $0.7834 \pm 0.0036$ | $0.9121 \pm 0.0097$ |
| **Zero Imputation w/ SN (ours.)** | $\mathbf{0.7106 \pm 0.0005}$ | $\mathbf{0.6911 \pm 0.0022}$ | $\mathbf{0.8096 \pm 0.0010}$ | $\mathbf{0.7914 \pm 0.0012}$ | $\mathbf{0.9283 \pm 0.0011}$ |
| Zero Imputation w/ BN | $0.7033 \pm 0.0035$ | $0.6691 \pm 0.0081$ | $0.7828 \pm 0.0136$ | $0.7749 \pm 0.0083$ | $0.9026 \pm 0.0105$ |
| Zero Imputation w/ LN | $0.7049 \pm 0.0035$ | $0.6811 \pm 0.0062$ | $0.7944 \pm 0.0015$ | $0.7820 \pm 0.0007$ | $0.9127 \pm 0.0056$ |
| Dropout | $0.7047 \pm 0.0021$ | $0.6747 \pm 0.0069$ | $0.7926 \pm 0.0049$ | $0.7802 \pm 0.0049$ | $0.9101 \pm 0.0054$ |
| Mean Imputation | $0.7054 \pm 0.0025$ | $0.6766 \pm 0.0067$ | $0.7913 \pm 0.0019$ | $0.7839 \pm 0.0021$ | $0.9117 \pm 0.0075$ |
| Median Imputation | $0.7056 \pm 0.0012$ | $0.6713 \pm 0.0058$ | $0.7865 \pm 0.0034$ | $0.7818 \pm 0.0014$ | $0.8975 \pm 0.0060$ |
| $k$-NN | $0.7052 \pm 0.0010$ | $\mathbf{0.6874 \pm 0.0045}$ | $0.8000 \pm 0.0050$ | $0.7843 \pm 0.0024$ | $0.9107 \pm 0.0075$ |
| MICE | $0.7075 \pm 0.0022$ | $\mathbf{0.6902 \pm 0.0058}$ | $0.7957 \pm 0.0037$ | $\mathbf{0.7875 \pm 0.0032}$ | $0.9224 \pm 0.0021$ |
| SoftImpute | $0.7014 \pm 0.0030$ | $\mathbf{0.6868 \pm 0.0040}$ | $0.7990 \pm 0.0050$ | $0.7828 \pm 0.0042$ | $0.9224 \pm 0.0019$ |
| GMMC | $0.7072 \pm 0.0016$ | $0.6816 \pm 0.0067$ | $0.7984 \pm 0.0062$ | $\mathbf{0.7884 \pm 0.0029}$ | $0.9109 \pm 0.0045$ |
| GAIN | $0.7065 \pm 0.0015$ | $0.6765 \pm 0.0060$ | $0.7956 \pm 0.0020$ | $0.7847 \pm 0.0031$ | $0.9091 \pm 0.0067$ |

the model with SN. Note that the same tendency has been observed regardless of test instances or datasets. This implies that models with SN yield more calibrated predictions; as more features are known for a particular instance, the variance of prediction for that instance should decrease (since we generated independent masks in Figure 1). It is also worthwhile to note that an AutoRec is a sigmoid-based network and Movielens datasets are known to have no MCAR hypothesis (Wang et al., 2018), in which Assumption 1 does not hold at all.

It is also validated that performance gains can be obtained by putting SN into AutoRec (Sedhain et al., 2015), CF-NADE (Zheng et al., 2016), and CF-UIcA (Du et al., 2018), which are the state-of-the-art among neural network based collaborative filtering models on several Movielens datasets (see Appendix B for detailed settings). In Table 1[3], we consider three different sized Movielens datasets. Note that AutoRec and CF-NADE allow two types of models according to data encoding (user- or item-rating vector based) and we consider both types. While we obtain performance improvements with SN in most cases, it is more prominent in user-rating based model in case of AutoRec and CF-NADE.

Furthermore, Table 2 compares our simply modification using SN on AutoRec and CF-UIcA with other state-of-the-art collaborative filtering models beyond neural networks. Unlike experiments of AutoRec in Table 1, which use the same network architectures proposed in original papers, here we could successfully learn more expressive network due to the stability obtained by using SN[4]. For Movielens 100K and 1M datasets, applying SN yields better or similar performance compared to other state-of-the-art collaborative filtering methods beyond neural network based models. It is important to note that all models outperforming AutoRec with SN on Movielens 10M are ensemble models while AutoRec with SN is a single model and it shows consistently competitive results across all datasets.

## 4.2 ELECTRONIC MEDICAL RECORDS (EMR) DATASETS

We further test VSP and SN for clinical time-series prediction with two Electronic Medical Records (EMR), namely PhysioNet Challenge 2012 (Silva et al., 2012) and the National Health Insurance Service (NHIS) datasets, which have intrinsic missingness since patients will only receive medical examinations that are considered to be necessary. We identify whether the VSP exists with the PhysioNet Challenge 2012 dataset (Silva et al., 2012). We randomly select one test point and plot in-hospital death probabilities as the number of examinations varies (Second column of Figure 1). Without SN, the in-hospital death probability increases as the number of examinations increases,

---

[3]We consider a CF-NADE without weight sharing and re-run experiments for fair comparisons because applying SN with weight sharing is not trivial. We also exclude averaging possible choices because it does not make big differences given unnecessary extra computational costs.

[4]Because overfitting is less with SN in AutoRec, we use twice the capacity than the existing AutoRec model.

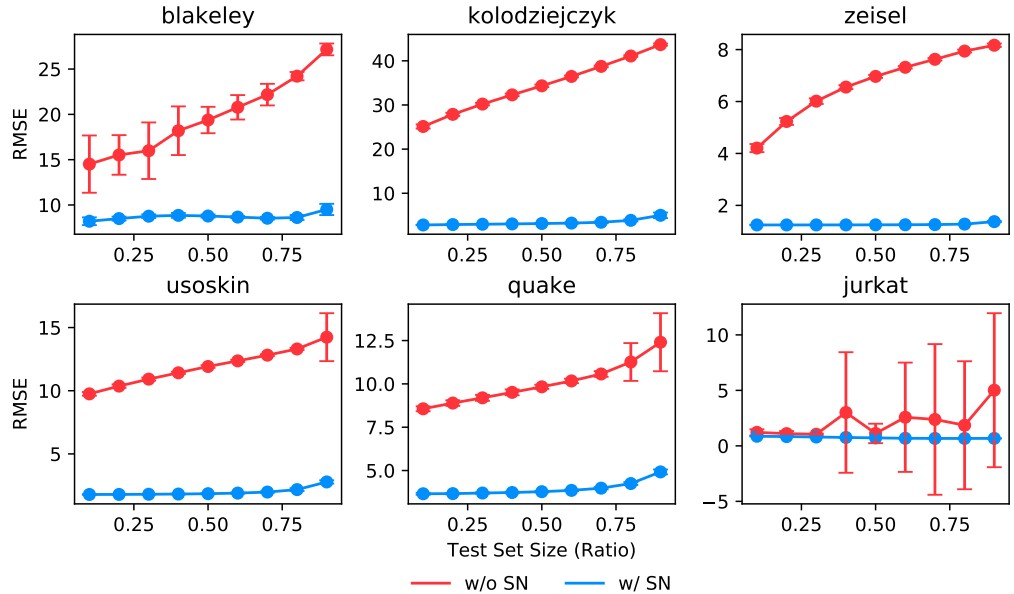

Figure 3: Debiasing variable sparsity using SN according to test set ratio on six imputation tasks of single cell RNA sequence dataset. Test RMSE with 95% confidence interval of 10-runs is provided.

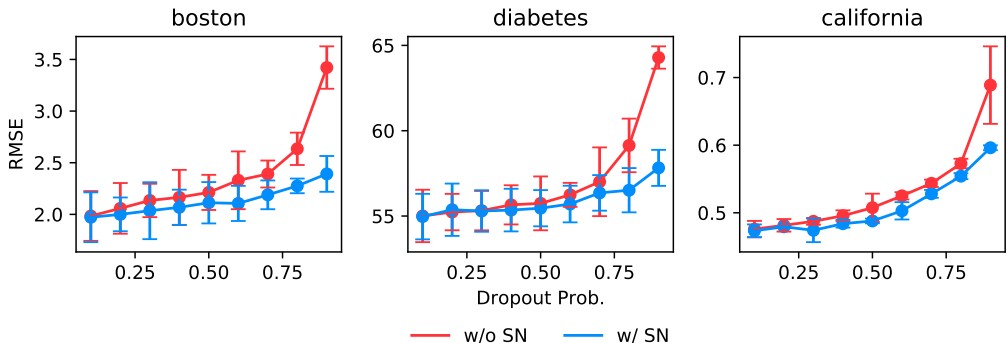

Figure 4: Debiasing variable sparsity using SN with respect to drop rates on three popular UCI regression datasets. Test RMSE with 95% confidence interval of 5-runs is provided.

even though there is no such tendency in the dataset statistics. However, SN corrects this bias so that in-hospital death probability is consistent regardless of the number of examinations. We observe a similar tendency for examples from the NHIS dataset as well.

Although SN corrects the VSP in both datasets, we perceive different behaviors in both cases in terms of actual performance changes. While SN significantly outperforms its counterpart without SN on NHIS dataset as shown in Table 3, it just performs similarly on PhysioNet dataset (results and detailed settings are deferred to Appendix C). However, SN is still valuable for its ability to prevent biased predictions in this mission-critical area.

In addition, we compare SN with other missing handling techniques to show the efficacy of SN although our main purpose is to provide a deeper understanding and the corresponding solution about the issue that the zero imputation, which is the simplest and most intuitive way of handling missing data, degrades the performance in training neural networks. Even with its simplicity, SN exhibits better or similar performances compared to other more complex techniques. Detailed descriptions of other missing handling techniques are deferred to Appendix H.

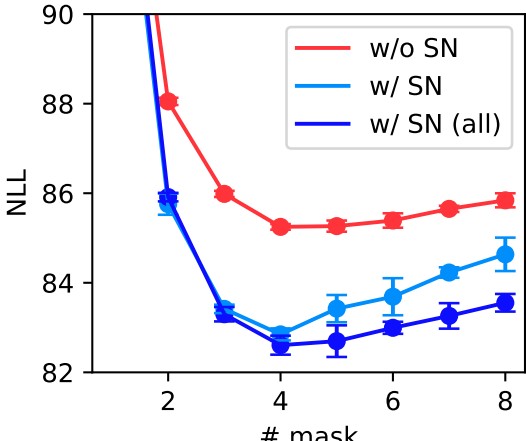

Figure 5: Negative log likelihood of MADE on binarized MNIST with and without SN.

### 4.3 SINGLE-CELL RNA SEQUENCE DATASETS

Single-cell RNA sequence datasets contain expression levels of specific RNA for each single cells. AutoImpute (Talwar et al., 2018) is one of the state-of-the-art methods that imputes missing data on single-cell RNA sequence datasets. We reproduce their experiments using authors' official implementation, and follow most of their experimental settings (see Appendix D for details).

As before, we first check whether VSP occurs in AutoImpute model. The third column in Figure 1 shows how the prediction of a AutoImpute model changes as the number of known entries changes. Although the number of RNAs found in the specific cell is less related to cell characteristics (upper right corner in Figure 1), the prediction increases as the number of RNAs found in the cell increases. This tendency is dramatically reduced with SN.

Figure 3 shows how imputation performance changes by being worked with SN to several single cell RNA sequence datasets with respect to the portion of train set (see Appendix D for more results). As we can see in Figure 3, SN significantly increases the imputation performance of AutoImpute model. In particular, the smaller the train data, the better the effect of SN in all datasets consistently. AutoImpute model is a sigmoid-based function and single cell RNA datasets (Talwar et al., 2018) do not have the MCAR hypothesis, unlike Assumption 1. Nevertheless, VSP occurs even here and it can be successfully alleviated by SN with huge performance gain. It implies that SN would work for other neural network based imputation techniques.

### 4.4 DROPOUT ON UCI DATASETS

While SN primarily targets to fix the VSP in the input layer, it could be also applied to any layers of deep neural networks to resolve VSP. The typical example of having heterogeneous sparsity in hidden layers is when we use dropout (Srivastava et al., 2014), which can be understood as another form of zero imputation but at the hidden layers; with Bernoulli dropout, the variance of the number of zero units across instances is $np(1-p)$ ($n$: the dimension of hidden layer, $p$: drop rate). While dropout partially handles this VSP issue by scaling $1/(1-p)$ in the training phase[5], SN can *exactly* correct VSP of hidden layers by considering individual level sparsity (Note that the scaling of dropout can be viewed as applying SN in an average sense: $E\left[\|\mathbf{m}\|_1\right] = n(1-p)$ and $K = n$).

---

[5]In almost all of the deep learning frameworks such as PyTorch, TensorFlow, Theano and Caffe, this inverted dropout is supported.

Figure 4 shows how the RMSE changes as the drop rate changes with and without SN on three popular UCI regression datasets (Boston Housing, Diabetes, and California Housing)[6]. As illustrated in Figure 4, the larger the drop rate, the greater the difference of RMSE between with and without SN. To explain this phenomenon, we define the degree of VSP as the inverse of *signal-to-noise ratio* with respect to the number of active units in hidden layers: $\sqrt{p/(n(1-p))}$ (expected number of active units over its standard deviation). As can be seen from the figure, the larger drop rate $p$ is, the more severe degree of VSP is and thus the greater protection by SN against performance degradation.

## 4.5 DENSITY ESTIMATION

In the current literature of estimating density based on deep models, inputs with missing features are in general not largely considered. However, we still may experience the VSP since the proportion of zeros in the data itself can vary greatly from instance to instance. In this experiment, we apply SN to MADE (Germain et al., 2015), which is one of the modern architectures in neural network-based density estimation. We reproduce binarized MNIST (LeCun, 1998) experiments of Germain et al. (2015) measuring negative log likelihood (the lower the better) of test dataset while increasing the number of masks. Figure 5 illustrates the effect of using SN. Note that MADE uses masks in the hidden layer that are designed to produce proper autoregressive conditional probabilities and variable sparsity arises across hidden nodes. SN can be trivially extended to handle this case as well and the corresponding result is given in the figure denoted as `w/SN(all)`[7]. We reaffirm that SN is effective even when MCAR assumption is not established.

## 5 RELATED WORKS

**Missing handling techniques**   Missing imputation can be understood as a technique to increase the generalization performance by injecting plausible noise into data. Noise injection using global statistics like mean or median values is the simplest way to do this (Lipton et al., 2016; Śmieja et al., 2018). However, it could lead to highly incorrect estimation since they do not take into consideration the characteristics of each data instance (Tresp et al., 1994; Che et al., 2018). To overcome this limitation, researchers have proposed various ways to model individualized noise using autoencoders (Pathak et al., 2016; Gondara & Wang, 2018), or GANs (Yoon et al., 2018; Li et al., 2019). However, those model based imputation techniques have not properly worked for high dimensional datasets with the large number of features and/or extremely high missing rates (Yoon et al., 2018) because excessive noise can ruin the training of neural networks rather increasing generalization performance.

For this reason, in the case of high dimensional datasets such as collaborative filtering or single cell RNA sequences, different methods of handling missing data have been proposed. A line of work simply used zero imputation by minimizing noise level and achieved state-of-the-art performance on their target datasets (Sedhain et al., 2015; Zheng et al., 2016; Talwar et al., 2018). In addition, methods using low-rank matrix factorization have been proposed to reduce the input dimension, but these methods not only cause lots of information loss but also fail to capture non-linearity of the input data (Hazan et al., 2015; Bachman et al., 2017; He et al., 2017). Vinyals et al. (2016); Monti et al. (2017) proposed recurrent neural network (RNN) based methods but computational costs for these methods are outrageous for high dimensional datasets. Also, it is not natural to use RNN-based models for non-sequential datasets.

---

[6]The most experimental settings are adopted from Klambauer et al. (2017); Littwin & Wolf (2018)'s UCI experiments (see Appendix E for detail).

[7]The detailed description of the extension is deferred to Appendix F.

**Other forms of Sparsity Normalization**   We discuss other forms of SN to alleviate VSP, already in use unwittingly due to empirical performance improvements. DropBlock (Ghiasi et al., 2018) compensates activation from dropped features by exactly counting mask vector similar to SN (similar approach discussed in Section 4.4.) It is remarkable that we can find models using SN-like normalization even in handling datasets without missing features. For example, in CBOW model (Mikolov et al., 2013) where the number of words used as an input depends on the position in the sentence, it was later revealed that SN like normalization has a practical performance improvement. As an another example, Kipf & Welling (2017) applied Laplacian normalization which is the standard way of representing a graph in graph theory, can naturally handle heterogeneous node degrees and precisely matches the SN operation. In this paper, we explicitly extend SN, which was limited and unintentionally applied to only few settings, to a model agnostic technique.

## 6   CONCLUSION

We identified *variable sparsity problem (VSP)* caused by zero imputation that has not been explicitly studied before. To best of our knowledge, this paper provided the first theoretical analysis on why zero imputation is harmful to inference of neural networks. We showed that variable sparsity problem actually exists in diverse real-world datasets. We also confirmed that theoretically inspired normalizing method, Sparsity Normalization, not only reduces the VSP but also improves the generalization performance and stability of feed-forwarding of neural networks with missing values, even in areas where existing missing imputation techniques do not cover well (e.g., collaborative filtering, single cell RNA datasets).

### ACKNOWLEDGMENTS

This work was supported by the Institute of Information & Communications Technology Planning & Evaluation (IITP) grants (No.2016-0-00563, No.2017-0-01779, and No.2019-0-01371), the National Research Foundation of Korea (NRF) grant funded by the Korea government (MSIT) grants (No.2018R1A5A1059921 and No.2019R1C1C1009192), the Samsung Research Funding & Incubation Center of Samsung Electronics via SRFC-IT1702-15, the National IT Industry Promotion Agency grant funded by the Ministry of Science and ICT, and the Ministry of Health and Welfare (NO. S0310-19-1001, Development Project of The Precision Medicine Hospital Information System (P-HIS)).

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

## A  PROOFS

### A.1  PROOF OF THEOREM 1

*Proof.* From the definition of $h_l, w_l^1, h_l^0, \tilde{h}_l^0, m_l$, the following equation holds.

$$E[h_l^1] = n_0 E[w_l^1 h_1^0] = n_0 E[w_l^1 \tilde{h}_l^0 m_l]$$

From the Assumption 1, $w_l^1, \tilde{h}_l^0$, and $m_l$ are independent of each other. Thus,

$$E[h_l^1] = n_0 E[w_l^1] E[\tilde{h}_l^0] E[m_l]$$

Similarly, the following holds.

$$E[h_l^i] = n_{i-1} E[w_l^i h_l^{i-1}] \text{ for } i = 1, \cdots, L$$

Since $h_l^{i-1}$ and $w_l^i$ are independent of each other by the Assumption 1 and the definition of $h_l^{i-1}$, $E[h_l^i] = n_{i-1} E[w_l^i] E[h_l^{i-1}]$. Therefore,

$$E[h_l^L] = \prod_{i=1}^{L} n_{i-1} E[w_l^i] E[\tilde{h}_l^0] E[m_l] = \prod_{i=1}^{L} n_{i-1} \mu_w^i \mu_x \mu_m$$

$\square$

### A.2  PROOF OF THEOREM 2

*Proof.* From the definition of $h_l, w_l^1, h_l^0, \tilde{h}_l^0, m_l$ and the property of an affine function $\sigma(E[\cdot]) = E[\sigma(\cdot)]$, the following equation holds.

$$E[h_l^1] = \sigma \left( n_0 E[w_l^1 h_l^0] + E[b_l^1] \right) = \sigma \left( n_0 E[w_l^1 \tilde{h}_l^0 m_l] + E[b_l^1] \right)$$

From the Assumption 1, $w_l^1, \tilde{h}_l^0$, and $m_l$ are independent of each other. Thus,

$$\begin{aligned} E[h_l^1] &= \sigma \left( n_0 E[w_l^1] E[\tilde{h}_l^0] E[m_l] + E[b_l^1] \right) \\ &= \sigma \left( n_0 \mu_w^1 \mu_x \mu_m + \mu_b^1 \right) \\ &= f_1(\mu_x \mu_m) \end{aligned}$$

Similarly, the following holds.

$$E[h_l^i] = \sigma \left( n_{i-1} E[w_l^i h_l^{i-1}] + E[b_l^i] \right) \text{ for } i = 1, \cdots, L$$

Since $h_l^{i-1}$ and $w_l^i$ are independent of each other by the Assumption 1 and the definition of $h_l^{i-1}$,

$$\begin{aligned} E[h_l^i] &= \sigma \left( n_{i-1} E[w_l^i] E[h_l^{i-1}] + E[b_l^i] \right) \\ &= \sigma \left( n_{i-1} \mu_w^i E[h_l^{i-1}] + \mu_b^i \right) \\ &= f_i(E[h_l^i]) \end{aligned}$$

Therefore,

$$E[h_l^L] = f_L \circ \cdots \circ f_1(\mu_x \mu_m)$$

$\square$

### A.3 Proof of Theorem 3

*Proof.* From the definition of $h_l, w_l^1, h_l^0, \tilde{h}_l^0, m_l$ and the property of a convex function $E[\sigma(\cdot)] \geq \sigma(E[\cdot])$, the following equation holds.

$$E[h_l^1] \geq \sigma\left(n_0 E[w_l^1 h_l^0] + E[b_l^1]\right) = \sigma\left(n_0 E[w_l^1 \tilde{h}_l^0 m_l] + E[b_l^1]\right)$$

From the Assumption 1, $w_l^1, \tilde{h}_l^0$, and $m_l$ are independent of each other. Thus,

$$\begin{aligned} E[h_l^1] &\geq \sigma\left(n_0 E[w_l^1] E[\tilde{h}_l^0] E[m_l] + E[b_l^1]\right) \\ &= \sigma\left(n_0 \mu_w^1 \mu_x \mu_m + \mu_b^1\right) \\ &= f_1(\mu_x \mu_m) \end{aligned}$$

Similarly, the following holds.

$$E[h_l^i] \geq \sigma\left(n_{i-1} E[w_l^i h_l^{i-1}] + E[b_l^i]\right) \text{ for } i = 1, \cdots, L$$

Since $h_l^{i-1}$ and $w_l^i$ are independent of each other by the Assumption 1 and the definition of $h_l^{i-1}$,

$$\begin{aligned} E[h_l^i] &\geq \sigma\left(n_{i-1} E[w_l^i] E[h_l^{i-1}] + E[b_l^i]\right) \\ &= \sigma\left(n_{i-1} \mu_w^i E[h_l^{i-1}] + \mu_b^i\right) \\ &= f_i(E[h_l^{i-1}]) \end{aligned}$$

Since we assume that $\sigma$ is non-decreasing, we finally get

$$E[h_l^L] \geq f_L \circ \cdots \circ f_1(\mu_x \mu_m)$$

$\square$

### A.4 Proof of Theorem 4

*Proof.* By theorem 2, $E[h_l^L] = f_L \circ \cdots \circ f_1(E[\mathbf{h}_{\text{SN}}^0])$. Since $E[\mathbf{h}_{\text{SN}}^0] = E[\mathbf{h}^0] \cdot K/\mu_m$,

$$E[h_l^L] = f_L \circ \cdots \circ f_1(\mu_x \mu_m \cdot K/\mu_m) = f_L \circ \cdots \circ f_1(\mu_x \cdot K)$$

$\square$

# B    COLLABORATIVE FILTERING (RECOMMENDATION) DATASETS

## B.1    DETAILED EXPERIMENTAL SETTINGS OF TABLE 1

This subsection describes the experimental settings of detailed collaborative filtering tasks in Section 4. As already mentioned, we follow the settings of AutoRec, CF-NADE, and CF-UIcA as much as possible. We perform each experiments by 5 times, and report mean and 95% confidence intervals. We randomly select 10% of the ratings of each datasets for the test set (Harper & Konstan, 2016). As the dataset is too small for Movielens 100K and 1M datasets, the confidence interval tends to be large by changing the dataset split. Hence, the same dataset split is used in each 5 experiments in Table 1.

**AutoRec (Sedhain et al., 2015)**    We use two layer AutoRec model with 500 hidden units. For fair comparisons, we tune the hyper-parameters for weight decay in all experiments to have only one significant digit, and use a learning rate of $10^{-3}$ except on Movielens 10M where we use a learning rate of $10^{-4}$. We use full batch on Movielens 100K and 1M, mini-batch (1000) on Movielens 10M. Besides, we use Adam optimizer instead of Resilient Propagation (RProp) unlike the AutoRec paper. The RProp optimizer shows fast convergence but can only be used in full batch scenario. It is not possible for 12GB of GPU memory to use full batch in training a large dataset such as Movielens 10M. Thus, we decide to use Adam optimizer rather than RProp optimizer. Fortunately, although the optimizer is changed to Adam, the prediction performance is not degraded in most cases. The experimental results of comparing both optimizers are summarized in Table 4.

Table 4: Comparison of Test RMSE between Adam and Resilient Propagation optimizer on Movielens 100K, 1M, and 10M datasets.

| Datasets | Movielens 100K | | Movielens 1M | | Movielens 10M | |
|---|---|---|---|---|---|---|
| input vector | item vector | user vector | item vector | user vector | item vector | user vector |
| RProp | 0.8861 | 0.9437 | 0.8358 | **0.8804** | 0.782[†] | **0.867**[†] |
| **Adam** | **0.8831** | **0.9343** | **0.8306** | 0.8832 | **0.7807** | 0.8859 |

[†] : Taken from Sedhain et al. (2015).

**CF-NADE (Zheng et al., 2016)**    We use two layer CF-NADE model with 500 hidden units. For fair comparisons, we tune the hyper-parameters for weight decay in all experiments to have only one significant digit, and use a learning rate[8] of 0.001. Also, we use mini-batch (512) just following CF-NADE. Although the CF-NADE used weight sharing and averaging possible choices in addition to weight decay, we report the results without weight sharing and averaging possible choices in Table 1 because it is not clear how to apply SN with weight sharing, and there is almost no performance gain with averaging possible choices despite of its high computational costs. Furthermore, we do not experiment on Movielens 10M with item vector encoding because the authors of CF-NADE did not provide the results for it due to the complexity of the model.

**CF-UIcA (Du et al., 2018)**    We use the authors' official code and the train/test dataset splits[9] for these experiments. Since CF-UIcA is a model that accepts both user and item vector as input, it is not necessary to consider two types of encoding as in AutoRec or CF-NADE. On the other hand, it is reasonable to take different $K$ values for user and item vector with SN. We treat $K$ as 66 for the user vector and 110 for the item vector. For models without SN, 0.0001 is used as the parameter $\lambda$ for weight decay as suggested by the CF-UIcA. It is natural to use different $\lambda$ values for SN because the optimal $\lambda$ must be changed along with SN. Hence, we use $\lambda = 0.0005$ for Movielens 100K and $\lambda = 0.00006$ for Movielens 1M in the case of SN. As in CF-NADE, we do not test Movielens 10M because the authors of CF-UIcA also did not report for it owing to its high computational cost.

---

[8]The CF-NADE uses learning rate $5 \times 10^{-4}$ for Movielens 10M, but we use $10^{-3}$ for fast convergence. Therefore, the results can be somewhat different from original paper.

[9]https://github.com/thu-ml/CF-UIcA.

## B.2 DETAILED EXPERIMENTAL SETTINGS OF TABLE 2

In comparison with other state-of-the-art models, we used 1000 hidden units in AutoRec (Sedhain et al., 2015) with SN. While Sedhain et al. (2015) claimed that they were able to achieve enough performance only with 500 hidden units, 500 hidden units did not achieve sufficient performance when applying SN. The Figure 6 plots the test RMSE, changing the number of hidden units for Movielens 100K and 1M. We can see that 600 units for Movielens 100K and 900 units for Movielens 1M are necessary for getting better performance. Obviously, as datasets become more complex and larger, we need more network capacity. Therefore, we decide to use two times larger network capacity (1000 hidden units) to get better performance. The number of hidden units can also be viewed and tuned as a hyper-parameter, and we have not tuned much for the number of hidden units. Note that, unlike Table 1, we report the results with five random splits for all datasets in Table 2 to compare fairly with other state-of-the-art methods.

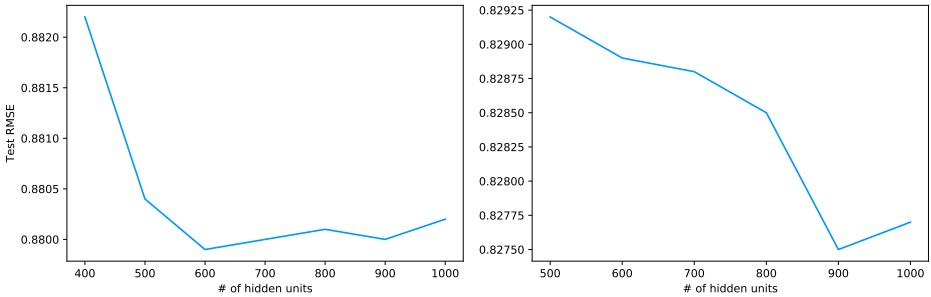

Figure 6: Test RMSE of AutoRec with SN on Movielens 100K **(left)** and Movielens 1M **(right)** with respect to the number of hidden units.

# C  ELECTRONIC MEDICAL RECORDS (EMR) DATASETS

## C.1  NHIS DATASET

Table 5: Debiasing variable sparsity in the case of applying dropout using SN on five disease identification tasks of NHIS dataset. Test AUROC with 95% confidence interval of 5-runs is provided.

| Dataset | Cardiovascular | Fatty Liver | Hypertension | Heart Failure | Diabetes |
|---|---|---|---|---|---|
| Zero Imputation w/o SN | $0.7084 \pm 0.0005$ | $0.6858 \pm 0.0065$ | $\mathbf{0.8023 \pm 0.0054}$ | $0.7876 \pm 0.0012$ | $\mathbf{0.9263 \pm 0.0026}$ |
| **Zero Imputation w/ SN (ours.)** | $\mathbf{0.7105 \pm 0.0009}$ | $\mathbf{0.6941 \pm 0.0011}$ | $\mathbf{0.8086 \pm 0.0016}$ | $\mathbf{0.7922 \pm 0.0015}$ | $\mathbf{0.9303 \pm 0.0029}$ |

The NHIS dataset, which is from National Health Insurance Service (NHIS), consists of medical diagnosis of around 300,000 people. The goal is to predict the occurrence of 5 diseases. Each patients takes 34 examinations over 5 years. We split dataset into two set (train and test), where the ratio of train and test split is 3:1. We pre-process input data with min-max normalization, which makes min and max values of each features be zero and one following GAIN (Yoon et al., 2018). We train 2 layer neural networks which have 50 and 30 hidden units each, and evaluate the model with AUROC. We use ReLU activation, and dropout rate as $0.8$ (if applied). Since these dataset is too imbalance, we apply class weight to the loss function for handling label imbalance. Besides, we use Adam optimizer with learning rate $10^{-2}$ without weight decay, and full batch. We evaluate the model on 5 times and report mean and 95% confidence interval. We also observe that Sparsity Normalization makes performance gain even when the dropout is integrated in the networks (See Table 5).

**Data source**  This study used the National Health Insurance System-National Health Screening Cohort (NHIS-HEALS)* data derived from a national health screening program and the national health insurance claim database in the National Health Insurance System (NHIS) of South Korea. Data from the NHIS-HEALS[10] was fully anonymized for all analyses and informed consent was not specifically obtained from each participant. This study was approved and exempt from informed consent by the Institutional Review Board of Yonsei University, Severance Hospital in Seoul, South Korea (IRB no.4-2016-0383).

**Data Availability**  Data cannot be shared publicly because of the provisions of the National Health Insurance Service (NHIS). Korean legal restrictions prohibit authors from making the data publicly available, and the authority implemented the restrictions is NHIS (National Health Insurance Service), one of the government agency of Republic of Korea. NHIS provides limited portion of anonymized data to the researchers for the purpose of the public interest. However, they exclusively provide data to whom made direct contact of the NHIS and agreed to policies of NHIS. Redistribution of the data is not permitted for the researchers. The contact name and the information to which the data request can be sent: Haeryoung Park Information analysis department Big data operation room NHISS `Tel:` `+82-33-736-2430`. E-mail: `lumen77@nhis.or.kr`.

---

[10]Seong SC, Kim YY, Park SK, et al. Cohort profile: the National Health Insurance Service-National Health Screening Cohort (NHIS-HEALS) in Korea. BMJ Open 2017;7:e016640. pmid:28947447.

## C.2   PHYSIONET CHALLENGE 2012 DATASET

Table 6: Debiasing variable sparsity using SN on mortality prediction task of PhysioNet Challenge 2012. Test AUROC with 95% confidence interval of 5-runs is provided.

| Model | Test AUROC |
|---|---|
| Zero Imputation w/o SN | **0.8177 ± 0.0071** |
| **Zero Imputation w/ SN (ours.)** | **0.8152 ± 0.0010** |

PhysioNet Challenge 2012 dataset (Silva et al., 2012) consists of 48 hours-long multivariate clinical time series data from intensive care unit (ICU). Most of the experimental settings are followed by BRITS (Cao et al., 2018). We divide 48 hours into 288 timesteps, which contain 35 examinations each. The goal of this task is to predict in-hospital death. We use dataset split given by PhysioNet Challenge 2012 (Silva et al., 2012) where each of them contains 4000 data points. In the preprocessing phase, we standardize (make mean as zero and standard deviation as one for each features) the input features and fill zero for missing values (zero imputation). We train single layer LSTM network which has 108 hidden units, and evaluate the model with AUROC. We apply class weight to handle imbalance problem in the dataset like in setting above. We use Adam optimizer with learning rate $2 \times 10^{-4}$, 512 batch size, and early stopping method based on AUROC of validation set. We evaluate the model on 5 times and report mean and 95% confidence interval. Note that input of LSTM model is the results for 35 medical examinations at a specific timestamp. Hence, we apply SN separately for each timestamp. As the aforementioned state of Section 4.2, SN could not make significant performance gain in PhysioNet Challenge 2012 dataset (Silva et al., 2012) though Sparsity Normalization eases the VSP (See Table 6). Nevertheless, SN is still valuable for its ability to prevent biased predictions in this mission-critical area.

## D SINGLE-CELL RNA SEQUENCE DATASETS

In the AutoImpute experiments, we run the experiments using the author's public code[11].Talwar et al. (2018) reported experimental results on eight datasets (Blakeley, Jurkat, Kolodziejczyk, Preimplantation, Quake, Usoskin, Zeisel, and PBMC). Since we can obtain preprocessed datasets only for seven datasets except PBMC, we run experiments on seven single cell RNA sequence datasets[12]. All experimental settings without SN are exactly followed by the author's code and the hyper-parameter settings published in Table 2 of their paper. For the model integrated with SN, all the experimental settings are followed by the original model except that the smaller threshold for early stopping is taken because the models with SN tends to be underfitted by using the threshold as suggested by the authors[13]. In addition, Talwar et al. (2018) conducted experiments only on cases with test set ratios of {0.1, 0.2, 0.3, 0.4, 0.5}, but we explored more test set ratios {0.1, 0.2, 0.3, 0.4, 0.5, 0.6, 0.7, 0.8, 0.9} to show that the SN works well under more sparsity (extremely high test set ratio). Like the authors' settings, we perform 10 experiments and report mean and 95% confidence intervals. We change the test set split on every trials. It is remarkable that although hyper-parameters are not favorable for SN, SN performs similarly or better on all seven datasets that we concerned (See Figure 3 and 7).

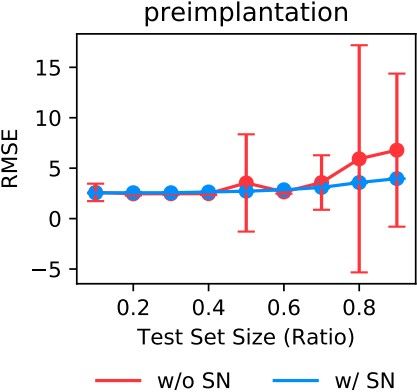

Figure 7: Debiasing variable sparsity using SN according to test set ratio on imputation task of Preimplantation dataset. Test RMSE with 95% confidence interval of 10-runs is provided.

## E DROPOUT ON UCI DATASETS

Most experimental settings are adopted from Klambauer et al. (2017); Littwin & Wolf (2018)'s UCI experiments. We use ReLU (Glorot et al., 2011) networks of 4 hidden layers with 256 units. We use Mean Square Error (MSE) for loss function and Adam optimizer without weight decay, $\epsilon = 10^{-8}$ and learning rate $10^{-4}$. The batch size used for training is 128. We split 10% from the dataset for test and use 20% of training set as the validation set. For all inputs, we applied min-max normalization, which makes min and max values of each features be 0 and 1. Note that, we use dense datasets without any missing attributes to focus on the effects of dropout. All the datasets are used as provided in the package in `sklearn.datasets`.

---

[11]`https://github.com/divyanshu-talwar/AutoImpute`

[12]`https://drive.google.com/drive/folders/1q2ho_cNfsQJNbdCt9j0nwlZv-Roj_yK1`

[13]We slightly tune the hyper-parameter $\lambda$ of weight decay only for the Preimplantation dataset with SN ($\lambda = 20$).

## F    DENSITY ESTIMATION

To reproduce binarized MNIST experiments of MADE (Germain et al., 2015), we adopt and slightly modify a public implementation of MADE[14]. Figure 5 shows our reproduction results of Figure 2 in the original paper. We follow most settings of the MADE. We use a single hidden layer MADE networks with 500 units, learning rate 0.005, and test on authors' binarized MNIST dataset[15]. The only difference from the original authors' implementation is that we used Adam ($\epsilon = 10^{-8}$, and no weight decay), rather using Adagrad. Because it is observed that the model is undefitted with Adagrad while the learning rates of each element of weight matrix is dropped so rapidly.

Since there is no missingness in the Binarized MNIST dataset, it cannot be divided by $\|M\|_1$ as suggested by Algorithm 1. In this experiment, we regard $\|M\|_1$ as $\|\mathbf{h}^0\|_0$ so as not to lose the generality. That is, all pixels that are 0 in the binarized MNIST dataset are regarded as missing. We label results of this with w/SN and plot them in Figure 5. As the aforementioned state in Section 4.5, MADE can also cause variable sparsity by mask matrices of each weight. In MADE, the connection between specific units is forcibly controlled through a mechanism of element-wise product of a specific mask matrix $M^i$ in weight $W^i \in \mathbb{R}^{n_i \times n_{i-1}}$. These mask matrices also cause variation in sparsity. We plot the results of using the new mask matrix $M_{\text{SN}}^i$ in place of the mask matrix $M^i$ used by MADE with w/SN(all) in Figure 5. The method of calculating the new mask matrix $M_{\text{SN}}^i$ using the existing mask matrix $M^i$ is as follows:

$$M_{\text{SN}}^i \leftarrow (\mathbf{1}^T M^i \mathbf{1}/n_i) \cdot M^i \oslash (M^i \mathbf{1}\mathbf{1}^T)$$

where $\oslash$ denotes element-wise division and $\mathbf{1}$ is a column vector where all elements are 1. $(\mathbf{1}^T M^i \mathbf{1}/n_i)$ corresponds to $K$ and $(M^i \mathbf{1}\mathbf{1}^T)$ does to $\|\mathbf{m}\|_1$ in Algorithm 1.

We demonstrate the effectiveness of SN even in situations where the learning rate is smaller. Smaller learning rate (0.001) shows the effect of SN more clearly despite of their longer training time as show in Figure 8.

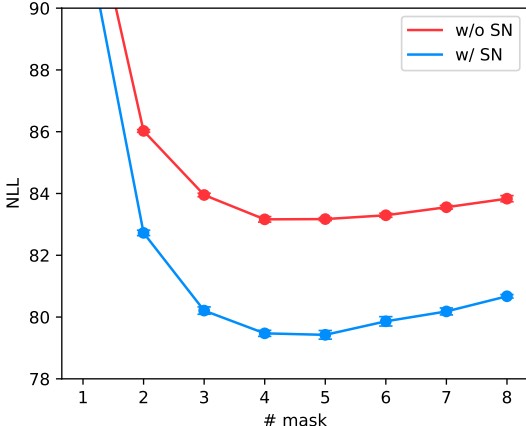

Figure 8: Negative log-likelihood of MADE on binarized MNIST with and without SN in the case of small learning rate (0.001).

## G    MACHINE DESCRIPTION

We perform all the experiments on a Titan X with 12GB of VRAM. 12 GB of VRAM is not always necessary, and most experiments require smaller VRAM.

---

[14]https://github.com/karpathy/pytorch-made
[15]https://github.com/mgermain/MADE/releases/download/ICML2015/binarized_mnist.npz

## H    COMPARISON TO OTHER MISSING HANDLING TECHNIQUES

In this section, we compare Sparsity Normalization with other missing handling techniques. Although the main contribution of our paper is to provide a deeper understanding and the corresponding solution about the issue that the zero imputation, the simplest and most intuitive way of handling missing data, degrades the performance in training neural networks. Nonetheless, we show that Sparsity Normalization is effective for high dimensional datasets (with a large number of features and high missing rates) via a collaborative filtering dataset, while showing that Sparsity Normalization has competitive results with other modern missing handling techniques for datasets with non-high dimensional setting (electronic medical records datasets, UCI datasets w/ and w/o MCAR assumption). For fair comparisons, we consider the tasks in Section 4, as well as the tasks used by modern missing handling techniques such as GAIN (Yoon et al., 2018) and Śmieja et al. (2018).

As baselines, we consider modern missing handling techniques such as GAIN, Śmieja et al. (2018) and their baselines as well.

- Zero Imputation w/o Sparsity Normalization: Missing values are replaced with zero.
- **Zero Imputation w/ Sparsity Normalization (ours.)**: Based on zero imputation, apply Sparsity Normalization (SN).
- Zero Imputation with Batch Normalization (Ioffe & Szegedy, 2015): Based on zero imputation, apply Batch Normalization (BN) only on the first layer.
- Zero Imputation with Layer Normalization (Lei Ba et al., 2016): Based on zero imputation, apply Layer Normalization (LN) only on the first layer.
- Dropout[16]: Missing values are replaced with zero and other values are divided by $\left( E_{(\mathbf{h}^0, \mathbf{m}) \in \mathcal{D}} [\|\mathbf{m}\|_1] / n_0 \right)$ like standard dropout (Srivastava et al., 2014). Dropout uses a single missing (drop) probability uniformly across all instances of the dataset while SN normalizes each data instance with its own missing rate.
- Mean Imputation: Missing values are replaced with the mean of those features.
- Median Imputation: Missing values are replaced with the median of those features.
- $k$-Nearest Neighbors ($k$-NN): Missing values are replaced with the mean of those features from $k$ nearest neighbor samples. We use $k = 5$ following Śmieja et al. (2018)'s experimental setting.
- Multivariate Imputation by Chained Equations (MICE): Proposed by Buuren & Groothuis-Oudshoorn (2010).
- SoftImpute (Mazumder et al., 2010)[17]
- Gaussian Mixture Model Compensator (GMMC): Proposed by Śmieja et al. (2018). In the case of GMMC, any activation functions except ReLU and RBF (Radial Basis Function) is prohibitive on the first hidden layer. Thus, the activation function of the first hidden layer is replaced by ReLU in all base architectures without ReLU.
- GAIN (Yoon et al., 2018)

We implement Mean Imputation, Median Imputation, MICE, $k$-NN and SoftImpute by python package `fancyimpute`. We use authors' official codes for GMMC[18] and GAIN[19]. Layer Normalization and Batch Normalization are not commonly considered in studies of handling missing data. However, we additionally take these as baselines because they have similarities to Sparsity Normalization in terms of stabilizing the statistics of a hidden layer[20] (See Appendix H.1.2 for deeper analysis).

---

[16]The GMMC (Śmieja et al., 2018) used this method as their baseline.

[17]In Yoon et al. (2018), this method was named Matrix.

[18]https://github.com/lstruski/Processing-of-missing-data-by-neural-networks

[19]https://github.com/jsyoon0823/GAIN

[20]We only consider LN and BN in the case of applying the first hidden layer. Because we find that the prediction performance is more worse when LN or BN applied to all the hidden layers, and it is difficult to fairly compare with the Sparsity Normalization.

## H.1 Collaborative Filtering (Recommendation) Dataset

In this section, we compare the SN with other missing handling techniques using the collaborative filtering dataset. Appendix H.1.1 compares the prediction performance of the baseline methods and the SN, while Appendix H.1.2 deeply analyzes the characteristics of the SN in comparison with Layer Normalization (Lei Ba et al., 2016) and Batch Normalization (Ioffe & Szegedy, 2015).

### H.1.1 Comparison of Prediction Performance

It is considered training an AutoRec (Sedhain et al., 2015) model on the Movielens 100K dataset. Most experimental settings are adopted from Section 4.1[21]. We evaluate each missing handling techniques on both data encoding (user- or item-rating vector) as shown in Table 7. In both encoding, Sparsity Normalization performs better or similar to other missing handling techniques. While some missing handling techniques perform poorly rather than zero imputation depending on the encoding, Sparsity Normalization improves performance consistently for both data encodings. It is worth mentioning that Sparsity Normalization performs statistically significantly better than all other baselines with item vector encoding, which is considered a better encoding scheme in most collaborative filtering models (Salakhutdinov et al., 2007; Sedhain et al., 2015; Zheng et al., 2016).

Table 7: Comparison between Sparsity Normalization and other missing handling techniques using AutoRec on Movielens 100K dataset. Test RMSE with 95% confidence interval of 5-runs is provided.

| Models | User vector | Item vector |
|---|---|---|
| Zero Imputation w/o Sparsity Normalization | $0.9346 \pm 0.0007$ | $0.8835 \pm 0.0003$ |
| **Zero Imputation w/ Sparsity Normalization (ours.)** | $\mathbf{0.9280 \pm 0.0023}$ | $\mathbf{0.8809 \pm 0.0011}$ |
| Zero Imputation w/ Batch Normalization | $0.9929 \pm 0.0088$ | $0.9205 \pm 0.0081$ |
| Zero Imputation w/ Layer Normalization | $0.9996 \pm 0.0131$ | $0.9396 \pm 0.0141$ |
| Dropout | $\mathbf{0.9252 \pm 0.0019}$ | $0.9268 \pm 0.0261$ |
| Mean Imputation | $0.9310 \pm 0.0017$ | $0.9206 \pm 0.0012$ |
| Median Imputation | $0.9333 \pm 0.0014$ | $0.9196 \pm 0.0017$ |
| $k$-NN | $0.9346 \pm 0.0007$ | $0.9133 \pm 0.0011$ |
| MICE (Buuren & Groothuis-Oudshoorn, 2010) | $0.9318 \pm 0.0015$ | $0.9209 \pm 0.0022$ |
| SoftImpute (Mazumder et al., 2010) | $\mathbf{0.9262 \pm 0.0003}$ | $0.8867 \pm 0.0007$ |
| GMMC (Śmieja et al., 2018) | $\mathbf{0.9331 \pm 0.0067}$ | $0.9109 \pm 0.0166$ |
| GAIN (Yoon et al., 2018) | $1.0470 \pm 0.0098$ | $1.0354 \pm 0.0101$ |

### H.1.2 Is Batch Normalization or Layer Normalization able to solve VSP?

Someone might wonder if Batch Normalization (Ioffe & Szegedy, 2015) or Layer Normalization (Lei Ba et al., 2016) have a similar effect to Sparsity Normalization by alleviating VSP. However, BN or LN can not solve VSP even though these three methods have something in common in terms of stabilizing the statistics of the hidden layer. To validate this, we compare SN with LN and BN by controlling the strength of weight decay regularization. We use the AutoRec model with Movielens 100K for these experiments as in Appendix H.1.1.

Figure 9 shows that VSP occurs in all cases except SN with weak regularization (left column). The model's prediction highly correlates the number of known entries in all methods except SN. On the other hand, strong regularization might seem to solve the VSP while the model shows relatively constant inference regardless of the number of known entries. However, the strong regularization is not an acceptable solution because it gives the less freedom of the model's inference making constant prediction (right column). It must not be natural that the predicted values of the model are almost constant regardless of input sample (we do not want a model that recommends the same movie no

---

[21]Only when applying Batch Normalization and Layer Normalization, we set the number of early stop iteration as $10,000$ (10 times of the other models) in order to prevent underfitting.

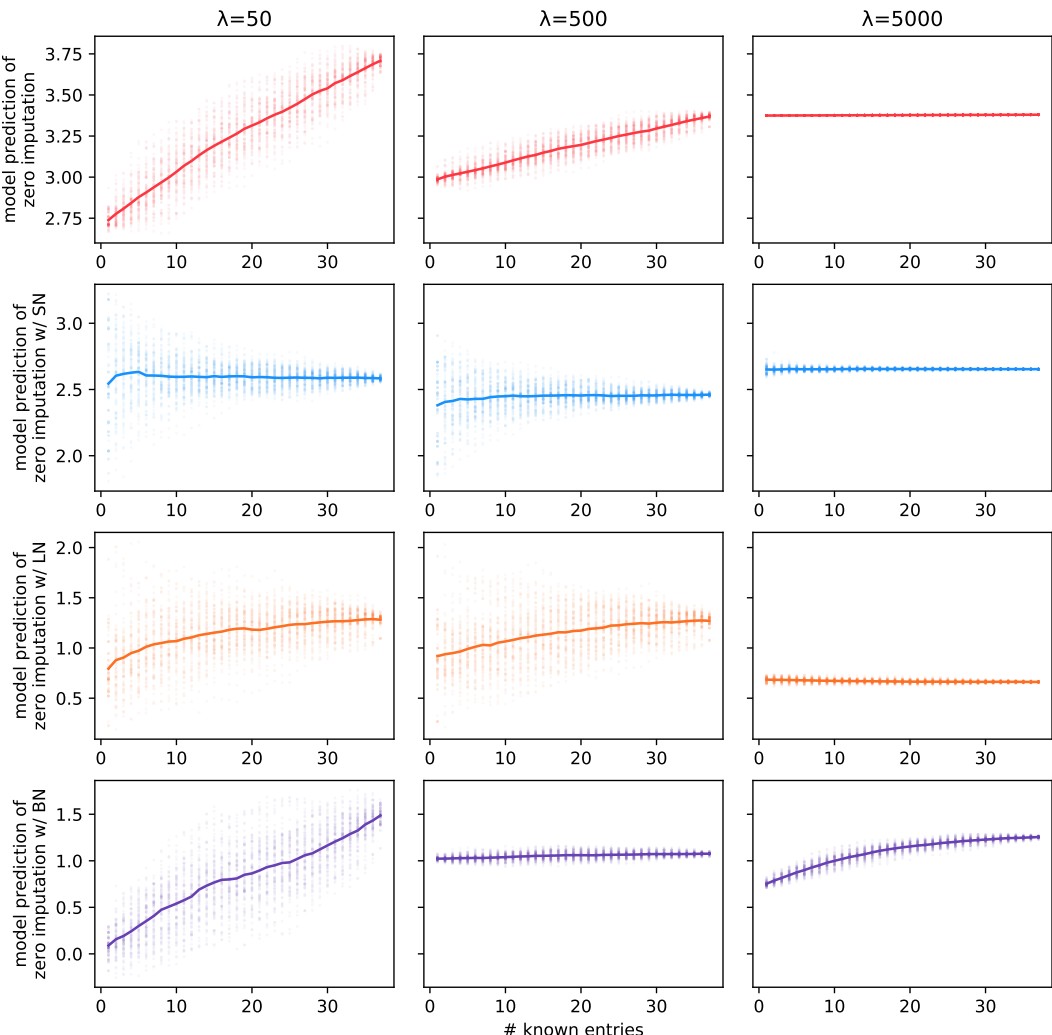

Figure 9: Predicted values of AutoRec (Sedhain et al., 2015) (user vector encoding) on Movielens 100K (collaborative filtering) dataset with zero imputation (w/ or w/o a normalization) according to the number of known entries for a randomly selected test point. Input masks are randomly sampled (to artificially control its sparsity level). For each target sparsity level through x-axis, 100 samples are drawn, scattering the predicted values and plotting the average in solid line. Predicted values of the model is also plotted according to the strength of the weight decay controlling $\lambda$ of the hyper-parameter for weight decay. **First row**: The vanilla zero imputation. **Second row (ours.)**: Zero imputation with Sparsity Normalization. **Third row**: Zero imputation with Layer Normalization (Lei Ba et al., 2016). **Fourth row**: Zero imputation with Batch Normalization (Ioffe & Szegedy, 2015). **First column**: Weak regularization ($\lambda = 50$). **Second column**: Moderate regularization ($\lambda = 500$). **Third column**: Strong regularization ($\lambda = 5000$).

matter which movies a user like/dislike!). This trend can also be seen in the process of tuning the hyper-parameter $\lambda$ for each model through that the optimal $\lambda$ value of each model except LN and BN is determined at around 500, whereas that of BN and LN is determined above 500000 (inordinate regularization). In other words, unlike SN, the VSP is not solved with LN or BN. Rather, strong regularization is able to solve the VSP, but this is not a direct solution to VSP forcing the model to choice constant predicted values irrespectively of the input. It is instructive note that the trend of Figure 9 is extremely consistent with the test points as Figure 1.

## H.2 ELECTRONIC MEDICAL RECORDS (EMR) DATASETS

We also compare Sparsity Normalization and baselines for the five disease identification tasks in the NHIS dataset used in Section 4.2. The results are described in Table 3. Sparsity Normalization shows better or similar performance compared to other baseline methods as well.

## H.3 UCI DATASETS

We further compare Sparsity Normalization with other missing handling techniques on UCI datasets which have relatively low missing rates and small feature dimension (non-high dimensional datasets). We consider the UCI datasets used in GAIN (Yoon et al., 2018) and GMMC (Śmieja et al., 2018). The datasets used in the both papers can be divided into two categories: missing features are intentionally injected (w/ MCAR assumption) or missing features exist inherently (w/o MCAR assumption).

We adopt the settings of Klambauer et al. (2017); Littwin & Wolf (2018)'s UCI exeperiments to use the same Multi Layer Perceptron (MLP) architecture as in Section 4.4: ReLU (Glorot et al., 2011) networks of 4 hidden layers with 256 units. The main purpose of imputation should be to improve prediction performance rather than imputation performance. In this reason, we just focus on the prediction performance of each missing handling techniques for UCI datasets. Because all UCI datasets used in this section are for imbalanced binary classification tasks, prediction performance is reported with AUROC rather than accuracy, and the class weight is considered in loss function. On top of that, we use Adam Optimizer in all experiments for fair comparison with baselines.

Though we adopt datasets used in GAIN (Yoon et al., 2018) and GMMC (Śmieja et al., 2018), we report quite different performance from that of the papers. Several possible reasons are as follows. First, GAIN and GMMC did not publish the train/test dataset split, thus we use our own split which is made under the similar settings of both papers. Second, MLP is used rather than logistic regression or Radial Basis Function Network (RBFN) which are used in GAIN and GMMC respectively. It is because we think that MLP is more reasonable and widely acceptable architecture nowadays[22] than the others. Furthermore, we use AUROC and class weights, unlike the GAIN and GMMC. The final possible reason is that we use Adam Optimizer for all models because SGD with learning rate decay is difficult for fair comparison when hyper-parameters are set in favor to a particular model. In these ways, we do our best to compare Sparsity Normalization and other missing handling techniques including GMMC and GAIN in the most fair and reasonable setting.

Table 8: Summary of UCI datasets with and without MCAR assumption.

| Dataset | w/ MCAR assumption | | | | | w/o MCAR assumption | | | | |
|---|---|---|---|---|---|---|---|---|---|---|
| | Breast | Spam | Credit | Crashes | Heart | Bands | Hepartitis | Horse | Mammographics | Pima |
| # Instances | 569 | 4601 | 30000 | 540 | 270 | 539 | 155 | 368 | 961 | 768 |
| # Attributes | 30 | 57 | 23 | 20 | 13 | 19 | 19 | 22 | 5 | 8 |
| Missing Rate (%) | 20.0 | 20.0 | 20.0 | 50.0 | 50.0 | 5.38 | 5.67 | 23.8 | 3.37 | 12.2 |

---

[22]Although there are MLP experiments on GMMC, they cover only one dataset and used too small capacity, also it was hard to get results similar to what the author reported in spite of running the author's public code.

### H.3.1 UCI DATASETS WITH MCAR ASSUMPTION

We deliberately inject missing values into the datasets which don't have any missing attributes internally (w/ MCAR assumption) to perform binary classification tasks. We consider Breast, Spam, and Credit datasets from GAIN (Yoon et al., 2018) and Crashes and Heart datasets from GMMC (Śmieja et al., 2018). We make 20% of all features be missing for the Breast, Spam, and Credit datasets following GAIN, and 50% for the Crashes and Heart datasets following GMMC. The summary of the datasets are described in Table 8. For Breast, Spam, and Credit datasets taken by GAIN, we perform min-max normalization which makes min and max values of each features be 0 and 1 following GAIN paper, and for Crashes and Heart datasets taken by GMMC, we perform another kind of min-max normalization which makes min and max values of each features be -1 and 1 following GMMC paper.

Table 9: Comparison between Sparsity Normalization and other missing handling techniques on five imbalanced binary classification tasks in UCI datasets with MCAR assumption. Test AUROC with 95% confidence interval of 5-runs is provided.

| Models | Breast | Spam | Credit | Crashes | Heart |
|---|---|---|---|---|---|
| Zero Imputation w/o SN | $0.9958 \pm 0.0067$ | $0.9699 \pm 0.0040$ | $0.7344 \pm 0.0135$ | $0.9070 \pm 0.1050$ | $0.8967 \pm 0.0268$ |
| **Zero Imputation w/ SN (ours.)** | $0.9992 \pm 0.0023$ | $0.9700 \pm 0.0032$ | $0.7292 \pm 0.0098$ | $0.8905 \pm 0.0934$ | $0.9055 \pm 0.0327$ |
| Zero Imputation w/ BN | $1.0000 \pm 0.0000$ | $0.9666 \pm 0.0051$ | $0.7178 \pm 0.0143$ | $0.9070 \pm 0.0645$ | $0.8857 \pm 0.0268$ |
| Zero Imputation w/ LN | $0.9979 \pm 0.0059$ | $0.9704 \pm 0.0031$ | $0.7353 \pm 0.0051$ | $0.9095 \pm 0.0972$ | $0.8863 \pm 0.0177$ |
| Dropout | $0.9897 \pm 0.0209$ | $0.9697 \pm 0.0031$ | $0.7313 \pm 0.0066$ | $0.9220 \pm 0.0553$ | $0.8857 \pm 0.0335$ |
| Mean Imputation | $0.9966 \pm 0.0089$ | $0.9690 \pm 0.0028$ | $0.7349 \pm 0.0066$ | $0.8990 \pm 0.1081$ | $0.9011 \pm 0.0522$ |
| Median Imputation | $0.9987 \pm 0.0045$ | $0.9695 \pm 0.0024$ | $0.7353 \pm 0.0044$ | $0.9080 \pm 0.0892$ | $0.9253 \pm 0.0407$ |
| $k$-NN | $1.0000 \pm 0.0000$ | $0.9613 \pm 0.0035$ | $0.5270 \pm 0.0171$ | $0.8940 \pm 0.0356$ | $0.8703 \pm 0.0435$ |
| MICE | $1.0000 \pm 0.0000$ | $0.9680 \pm 0.0041$ | $0.5315 \pm 0.0187$ | $0.9760 \pm 0.0424$ | $0.9066 \pm 0.0410$ |
| SoftImpute | $1.0000 \pm 0.0000$ | $0.9725 \pm 0.0021$ | $0.5375 \pm 0.0184$ | $0.9390 \pm 0.0754$ | $0.8846 \pm 0.0437$ |
| GMMC | $0.9987 \pm 0.0026$ | $0.9679 \pm 0.0029$ | $0.7355 \pm 0.0053$ | $0.9400 \pm 0.0422$ | $0.9264 \pm 0.0223$ |
| GAIN | $1.0000 \pm 0.0000$ | $0.9688 \pm 0.0029$ | $0.7356 \pm 0.0045$ | $0.9650 \pm 0.0360$ | $0.8879 \pm 0.0196$ |

The experimental results are summarized in Table 9. It is difficult to find a significant difference in prediction performance among each missing handling techniques for datasets with small feature dimensions. The results of these experiments are also consistent with experiments of GAIN (Yoon et al., 2018). Though the GAIN showed significantly better imputation performance compared to their baseline methods, prediction performances were not statistically significant (See Table 3 of the GAIN paper, and note that they didn't report 95% confidence interval but standard deviation). From these overall results, we conclude that SN is quite comparable for the datasets of low dimension/missing rate with MCAR assumption.

### H.3.2 UCI DATASETS WITHOUT MCAR ASSUMPTION

We compare SN and each missing handling techniques on the datasets that have internal missingness (w/o MCAR assumption). We consider Bands, Hepartitis, Horse, Mammographics, and Pima datasets experimented in the GMMC (Śmieja et al., 2018) paper (See Table 8 for statistics of the datasets). Following the GMMC paper, the min-max normalization is performed, which makes min and max values of each features be -1 and 1. As shown in Table 10, it is concluded that even without MCAR assumption, SN shows comparable results for the datasets of low dimension/missing rate.

Table 10: Comparison between Sparsity Normalization and other missing handling techniques on five imbalanced binary classification tasks in UCI datasets without MCAR assumption. Test AUROC with 95% confidence interval of 5-runs is provided.

| Models | Bands | Hepartitis | Horse | Mammographics | Pima |
|---|---|---|---|---|---|
| Zero Imputation w/o SN | $0.7851 \pm 0.0962$ | $0.8222 \pm 0.0826$ | $0.9090 \pm 0.0331$ | $0.8835 \pm 0.0083$ | $0.8466 \pm 0.0180$ |
| **Zero Imputation w/ SN (ours.)** | $0.7939 \pm 0.1011$ | $0.8556 \pm 0.1300$ | $0.9238 \pm 0.0311$ | $0.8787 \pm 0.0063$ | $0.8355 \pm 0.0233$ |
| Zero Imputation w/ BN | $0.7512 \pm 0.0716$ | $0.7889 \pm 0.1061$ | $0.8827 \pm 0.0362$ | $0.8933 \pm 0.0055$ | $0.8614 \pm 0.0106$ |
| Zero Imputation w/ LN | $0.8000 \pm 0.0789$ | $0.8222 \pm 0.0911$ | $0.9115 \pm 0.0404$ | $0.8854 \pm 0.0097$ | $0.7972 \pm 0.0074$ |
| Dropout | $0.7956 \pm 0.1012$ | $0.8194 \pm 0.1089$ | $0.9034 \pm 0.0507$ | $0.8816 \pm 0.0059$ | $0.8408 \pm 0.0292$ |
| Mean Imputation | $0.7765 \pm 0.0504$ | $0.8556 \pm 0.0895$ | $0.9009 \pm 0.0271$ | $0.8834 \pm 0.0113$ | $0.8382 \pm 0.0142$ |
| Median Imputation | $0.7997 \pm 0.0682$ | $0.8889 \pm 0.0544$ | $0.9133 \pm 0.0379$ | $0.8783 \pm 0.0070$ | $0.8334 \pm 0.0099$ |
| $k$-NN | $0.7571 \pm 0.0430$ | $0.8833 \pm 0.1047$ | $0.9300 \pm 0.0363$ | $0.8840 \pm 0.0141$ | $0.6237 \pm 0.0399$ |
| MICE | $0.7618 \pm 0.0297$ | $0.8333 \pm 0.0667$ | $0.9195 \pm 0.0431$ | $0.8814 \pm 0.0087$ | $0.6477 \pm 0.0655$ |
| SoftImpute | $0.7644 \pm 0.0475$ | $0.8139 \pm 0.1047$ | $0.9232 \pm 0.0124$ | $0.8848 \pm 0.0089$ | $0.5912 \pm 0.0341$ |
| GMMC | $0.7985 \pm 0.0605$ | $0.8500 \pm 0.0621$ | $0.9000 \pm 0.0124$ | $0.8844 \pm 0.0090$ | $0.8391 \pm 0.0201$ |
| GAIN | $0.7993 \pm 0.0785$ | $0.8222 \pm 0.0826$ | $0.8941 \pm 0.0322$ | $0.8859 \pm 0.0091$ | $0.8457 \pm 0.0178$ |

### H.4 CONCLUSION

In conclusion, Sparsity Normalization is significantly superior to other missing handling techniques for the high dimensional/missing rate datasets. Sparsity Normalization performs well compared to other modern missing handling techniques even on non-high dimensional datasets. Sparsity Normalization is valuable in that it performs better than other models and does not require additional training or parameters. Moreover, Sparsity Normalization is computationally inexpensive compared to Mean or Median Imputation because sparse tensors can be used to save computational costs to calculate first hidden layer not with mean or median imputation but with zero imputation (w/ or w/o SN). The reduced computational cost by using sparse tensors is relatively higher when we deal with high-dimensional datasets.

