# OpenReview forum: "Why Not to Use Zero Imputation? Correcting Sparsity Bias in Training Neural Networks"
_ICLR.cc/2020/Conference — Accept (Poster)_

### Official Review · AnonReviewer1 · 2019-10-20
**Official Blind Review #1**

**Rating:** 6

**Review:**

Zero imputation is studied from a different view by investigating its impact on prediction variability and a normalization scheme is proposed to alleviate this variation. This normalization scales the input neural network so that the output would not be affected much. While such simple yet helpful algorithms are plausible there are number of remaining issues:
1-	Zero imputation, as authors mentioned, is not an acceptable algorithm for imputation and improving on that via the normalization proposed in the paper cannot be counted as an exciting move in this area unless an extensive comparison shows it’s benefits over the many other existing techniques. I am interested to see how would the results be if you compare this simple algorithm with more complicated ones like GAIN or MisGAN. It is argued in the paper that with high dimensional data, your algorithm is more acceptable, but how would it be with in other cases?
2-	Your algorithm is only explained with neural net framework, how can we extend it to the other machine learning models?
3-	Is batch normalization used in your experiments? Scaling the activation in one layer to reduce its impact on the next layer is somehow similar to what happens in batch normalization, and I am wondering if BN makes any similar effect?
4-	Please provide labels for the x-axes in the figures.

------------------------------------------
After rebuttal:
Thanks for adding the extra experiments.
Looking at Table 9 in appendix, I am bit surprised to see that sometimes mean imputation works better than MICE (GAIN usually works good with large data). Maybe it attributes to the missing features. How did you choose to apply 20% missingness? randomly?

**Experience Assessment:**

I have read many papers in this area.

**Review Assessment: Checking Correctness Of Derivations And Theory:**

I assessed the sensibility of the derivations and theory.

**Review Assessment: Checking Correctness Of Experiments:**

I did not assess the experiments.

**Review Assessment: Thoroughness In Paper Reading:**

I read the paper at least twice and used my best judgement in assessing the paper.

---

> ### Author Response · Authors · 2019-11-13
> **Response to Reviewer #1 [Part 4/4]**
>
>
> [4. Please provide labels for the x-axes in the figures.]
> - We apologize for the confusion. We fixed this issue on our revised paper.

---

> ### Author Response · Authors · 2019-11-13
> **Response to Reviewer #1 [Part 3/4]**
>
> [3. Does Batch Normalization make similar effect to SN?]
>
> - We thank for the interesting suggestion. We performed experimental comparison against both BN and Layer Normalization (LN) (since BN could stabilize statistics of hidden layer but it does not consider instance wise characteristics). Please see Appendix H for the results and discussions.
>
> In our new experiments, SN significantly outperforms both LN and BN in most cases (or yields at least comparable performance in all cases).  Note that, in certain settings, LN and BN perform even worse than vanilla zero imputation with a large margin. For instance, on Movielens 100K (AutoRec, item vector encoding), RMSE of vanilla zero imputation, LN, BN as follows: 0.8835 ± 0.0003 (zero imputation w/o SN) vs. 0.9396 ± 0.0141 (LN) vs. 0.9205 ± 0.0081 (BN). Thus neither BN or LN seem to be effective in solving the VSP problem as SN.
>
> In all our previous experiments, we inadvertently did not consider Batch Normalization (BN) simply because BN is not widely used in dealing with tabular datasets despite its universality on vision tasks.

---

> > ### Author Response · Authors · 2019-11-15
> > **Additional Comment of Response to Reviewer #1 [Part 3/4]**
> >
> > We additionally investigate how the model outputs change with the number of known entries of the input when each normalization is applied (as we show in Figure 1). Since BN and LN do not specifically target VSP in the first place, they do not completely solve the bias caused by input sparsity levels. In particular, this phenomenon becomes more pronounced when L2 weight decay is not applied (or weight decay parameter lambda is small), hence we have large absolute values of weights. The details can be found in Appendix H.1.2.

---

> ### Author Response · Authors · 2019-11-13
> **Response to Reviewer #1 [Part 2/4]**
>
> [2. Your algorithm is only explained with neural net framework, how can we extend it to the other machine learning models?]
>
> - In this paper, we analyze the variable sparsity problem with the focus of neural networks. Our theoretical analysis can be seamlessly applied to certain non-neural network models (e.g. shallow linear regression is a special case of our analysis without hidden layers). However,  we need further research to confirm whether VSP occurs for all machine learning models in general.
>
> We believe that VSP for each model needs to be studied separately. Even within the neural network framework we are focusing on, there are different results depending on the type of activation functions and other details as you can see in Theorem 1-3.

---

> ### Author Response · Authors · 2019-11-13
> **Response to Reviewer #1 [Part 1/4]**
>
> We thank the reviewer for thoughtful and constructive feedback.
>
> [1. Comparison with other missing handling techniques]
> - First of all, the main contribution of our paper is to provide a deeper understanding and the corresponding solution about the issue that the zero imputation, the simplest and most intuitive way of handling missing data, degrades the performance in training neural networks. Hence, we only considered the vanilla zero imputation as our baseline in the submission since we do not claim that our corrected zero imputation (with SN) is the best for all tasks.
>
> However, some reviewers wanted to see direct comparisons against other state-of-the-art imputation techniques such as GAIN [1] and GMMC [2]. Hence, we performed additional comparisons against them on tasks considered in our paper as well as available tasks in [1] and [2].  Interestingly (and thanks to the reviewers who raised this issue), our corrected zero imputation (with SN), even with its simplicity, shows at least comparable or significantly better performances over all baselines, on all considered tasks. Here only two cases (Movielens 100K for high dimensional/missing rate case; NHIS dataset for low dimensional/missing rate case) are shown as examples and the rest are described in Appendix H:
>
> (Movielens 100K using item vector encoding)
> ----------------------------------------------------------------
>                   Model                  |         Test RMSE
> ----------------------------------------------------------------
>   Zero Imputation w/o SN   |     0.8835 ± 0.0003
>   Zero Imputation w/ SN     |     0.8809 ± 0.0011
> ----------------------------------------------------------------
>   Zero Imputation w/ BN     |     0.9205 ± 0.0081
>   Zero Imputation w/ LN     |     0.9396 ± 0.0141
>   Dropout                               |     0.9268 ± 0.0261
>   Mean Imputation              |     0.9206 ± 0.0012
>   Median Imputation           |     0.9196 ± 0.0017
>   kNN                                     |     0.9133 ± 0.0011
>   MiCE                                    |     0.9209 ± 0.0022
>   SoftImpute                        |     0.8867 ± 0.0007
>   GMMC                                |     0.9109 ± 0.0166
>   GAIN                                   |     1.0354 ± 0.0101
> ----------------------------------------------------------------
> (NHIS dataset diabetes identification task)
> ----------------------------------------------------------------
>                   Model                  |         Test AUROC
> ----------------------------------------------------------------
>   Zero Imputation w/o SN   |     0.9121 ± 0.0097
>   Zero Imputation w/ SN     |     0.9283 ± 0.0011
> ----------------------------------------------------------------
>   Zero Imputation w/ BN     |     0.9026 ± 0.0105
>   Zero Imputation w/ LN     |     0.9127 ± 0.0056
>   Dropout                               |     0.9101 ± 0.0054
>   Mean Imputation              |     0.9117 ± 0.0075
>   Median Imputation           |     0.8975 ± 0.0060
>   kNN                                     |     0.9107 ± 0.0075
>   MiCE                                    |     0.9224 ± 0.0021
>   SoftImpute                        |     0.9224 ± 0.0019
>   GMMC                                |     0.9109 ± 0.0045
>   GAIN                                   |     0.9091 ± 0.0067
> ----------------------------------------------------------------
> Note that the evaluation metrics are different for above two cases (RMSE for Movielens and AUROC for NHIS). In the most low dimensional and low missing rate cases, the problem is relatively easy, so all imputation methods work comparably well.
>
> We still believe that each imputation technique has its own advantages and disadvantages, and we do not claim that our corrected zero imputation (with SN) is always the best. However, we do believe that these new experiments show SN is a sufficiently competitive technique.
>
> [1] Jinsung Yoon, James Jordon, and Mihaela Van Der Schaar.  Gain: Missing data imputation using generative adversarial nets.  In Proceedings of the 35th International Conference on Machine Learning-Volume 71, 2018
> [2] Marek ́Smieja, Łukasz Struski, Jacek Tabor, Bartosz Zieli ́nski, and Przemysław Spurek. Processing of missing data by neural networks. In Advances in Neural Information Processing Systems, pp.2719–2729, 2018.

---

### Official Review · AnonReviewer3 · 2019-10-21
**Official Blind Review #3**

**Rating:** 6

**Review:**

This paper provides a novel solution to the variable sparsity problem, where the output of neural networks biased with respect to the number of missing inputs. The authors proposed a sparsity normalization algorithm to process the input vectors to encounter the bias. In experiments, the authors evaluated the proposed sparsity normalization model on multiple datasets: collaborative filtering datasets, electric medical records datasets, single-cell RNA sequence datasets and UCI datasets. Results show that the proposed normalization method improves the prediction performance and the predicted values of the neural network is more uniformly distributed according to the number of missing entries.

The paper describes a clear and specific machine learning problem. Then the authors demonstrate a simple normalization strategy is capable of fixing the issue of biased prediction. The paper has a well-organized structure to convey the motivation. Therefore, my opinion on this paper leans to an acceptation. My questions are mainly on the experiment section:

1) As shown in Table 2, there are various new collaborative filtering methods proposed after 2015, why the authors chose to extend AutoRec (Sedhain et al., 2015) but not other new methods?

2) In the experiments, you compare your model with zero imputation (Please correct me if w/o SN is not zero imputation). However, I think it is a common practice in machine learning that we perform imputation with mean or median values. I'm interested in knowing whether filling with mean/median values work with these datasets.

3) In section 4.5, you mentioned that "SN is effective
even when MCAR assumption is not established". However, I'm still not clear about the reason. I believe many machine learning datasets have NMAR (not missing at random) type of missing data, but not MCAR. So this is an important issue for me.

4) Does your model assume all input values are numerical but not categorical?

**Experience Assessment:**

I do not know much about this area.

**Review Assessment: Checking Correctness Of Derivations And Theory:**

I assessed the sensibility of the derivations and theory.

**Review Assessment: Checking Correctness Of Experiments:**

I assessed the sensibility of the experiments.

**Review Assessment: Thoroughness In Paper Reading:**

I read the paper thoroughly.

---

> ### Author Response · Authors · 2019-11-13
> **Response to Reviewer #3 [Part 4/4]**
>
>
> [4. Does your model assume all input values are numerical but not categorical?]
> - There is no restriction about the type of inputs in our analysis and the construction of our algorithm. In fact, CF-NADE and CF-UIcA for collaborative filtering datasets in our experiments, only allow categorical values for their inputs where SN successfully achieves the performance improvement. Another example of using SN for categorical input is density estimation tasks (binarized MNIST) in Section 4.5.

---

> > ### Comment · AnonReviewer3 · 2019-11-14
> > **Official Blind Review #3**
> >
> > Thank you for the responses. Adding the comparison with multiple recent imputation methods will surely significantly improve the impact of the paper.

---

> ### Author Response · Authors · 2019-11-13
> **Response to Reviewer #3 [Part 3/4]**
>
>
> [3. MCAR assumption]
> - We also fully agree that MCAR assumption is the one that are generally not well established in real cases. But, this assumption drastically simplifies our statements (as we know, theoretical analysis always requires some simplified assumptions and hence there's some gap with reality). Without this assumption, we can't get such neat statements since we have to worry about some worst cases in our analysis, but this does not mean SN is ineffective even in theory; we can still see that SN can reduce the dependency on sparsity level to some extent, although in a much more complex form. In order to make up for having such a simplified assumption, we experimentally show that variable sparsity problems actually exist in various real-world datasets even where the MCAR assumption does not hold, and that SN can relieve this problem.

---

> ### Author Response · Authors · 2019-11-13
> **Response to Reviewer #3 [Part 2/4]**
>
>
> [2. Comparison with other missing handling techniques]
> - First of all, the main contribution of our paper is to provide a deeper understanding and the corresponding solution about the issue that the zero imputation, the simplest and most intuitive way of handling missing data, degrades the performance in training neural networks. Hence, we only considered the vanilla zero imputation as our baseline in the submission since we do not claim that our corrected zero imputation (with SN) is the best for all tasks.
>
> However, some reviewers wanted to see direct comparisons against other state-of-the-art imputation techniques such as GAIN [1] and GMMC [2]. Hence, we performed additional comparisons against them on tasks considered in our paper as well as available tasks in [1] and [2].  Interestingly (and thanks to the reviewers who raised this issue), our corrected zero imputation (with SN), even with its simplicity, shows at least comparable or significantly better performances over all baselines, on all considered tasks. Here only two cases (Movielens 100K for high dimensional/missing rate case; NHIS dataset for low dimensional/missing rate case) are shown as examples and the rest are described in Appendix H:
>
> (Movielens 100K using item vector encoding)
> ----------------------------------------------------------------
>                   Model                  |         Test RMSE
> ----------------------------------------------------------------
>   Zero Imputation w/o SN   |     0.8835 ± 0.0003
>   Zero Imputation w/ SN     |     0.8809 ± 0.0011
> ----------------------------------------------------------------
>   Zero Imputation w/ BN     |     0.9205 ± 0.0081
>   Zero Imputation w/ LN     |     0.9396 ± 0.0141
>   Dropout                               |     0.9268 ± 0.0261
>   Mean Imputation              |     0.9206 ± 0.0012
>   Median Imputation           |     0.9196 ± 0.0017
>   kNN                                     |     0.9133 ± 0.0011
>   MiCE                                    |     0.9209 ± 0.0022
>   SoftImpute                        |     0.8867 ± 0.0007
>   GMMC                                |     0.9109 ± 0.0166
>   GAIN                                   |     1.0354 ± 0.0101
> ----------------------------------------------------------------
> (NHIS dataset diabetes identification task)
> ----------------------------------------------------------------
>                   Model                  |         Test AUROC
> ----------------------------------------------------------------
>   Zero Imputation w/o SN   |     0.9121 ± 0.0097
>   Zero Imputation w/ SN     |     0.9283 ± 0.0011
> ----------------------------------------------------------------
>   Zero Imputation w/ BN     |     0.9026 ± 0.0105
>   Zero Imputation w/ LN     |     0.9127 ± 0.0056
>   Dropout                               |     0.9101 ± 0.0054
>   Mean Imputation              |     0.9117 ± 0.0075
>   Median Imputation           |     0.8975 ± 0.0060
>   kNN                                     |     0.9107 ± 0.0075
>   MiCE                                    |     0.9224 ± 0.0021
>   SoftImpute                        |     0.9224 ± 0.0019
>   GMMC                                |     0.9109 ± 0.0045
>   GAIN                                   |     0.9091 ± 0.0067
> ----------------------------------------------------------------
> Note that the evaluation metrics are different for above two cases (RMSE for Movielens and AUROC for NHIS). In the most low dimensional and low missing rate cases, the problem is relatively easy, so all imputation methods work comparably well.
>
> We still believe that each imputation technique has its own advantages and disadvantages, and we do not claim that our corrected zero imputation (with SN) is always the best. However, we do believe that these new experiments show SN is a sufficiently competitive technique.
>
> [1] Jinsung Yoon, James Jordon, and Mihaela Van Der Schaar.  Gain: Missing data imputation using generative adversarial nets.  In Proceedings of the 35th International Conference on Machine Learning-Volume 71, 2018
> [2] Marek ́Smieja, Łukasz Struski, Jacek Tabor, Bartosz Zieli ́nski, and Przemysław Spurek. Processing of missing data by neural networks. In Advances in Neural Information Processing Systems, pp.2719–2729, 2018.

---

> ### Author Response · Authors · 2019-11-13
> **Response to Reviewer #3 [Part 1/4]**
>
> We thank the reviewer for thoughtful and constructive feedback.
>
> [1. Powerful backbone architecture on collaborative filtering datasets]
> - We use AutoRec (Sedhain et al., 2015) and its variant CF-NADE (Zheng et al., 2016) without any intention simply because many modern nn-based models are in fact variants of AutoRec. But, following the reviewers’ valuable suggestion, we consider CF-UIcA [3], one of the current state-of-the-arts, as a new backbone on collaborative filtering datasets, and consistently achieve even stronger performances (in terms of RMSE):
>     - Movielens 100K: 0.8945 ± 0.0024 (w/o SN) vs. 0.8793 ± 0.0017 (w/ SN)
>     - Movielens 1M: 0.8223 ± 0.0016 (w/o SN) vs. 0.8178 ± 0.0007 (w/ SN)
>
> Note that we do not test for Movielens 10M because the authors of CF-UIcA did not provide the results for it due to the complexity of the model.
>
> [3] Du, C., Li, C., Zheng, Y., Zhu, J., & Zhang, B. (2018, April). Collaborative filtering with user-item co-autoregressive models. In Thirty-Second AAAI Conference on Artificial Intelligence.

---

### Official Review · AnonReviewer2 · 2019-10-23
**Official Blind Review #2**

**Rating:** 6

**Review:**

This paper studies a very interesting phenomena in machine learning called VSP, that is the output of the model is highly affected via the level of missing values in its input.  The authors demonstrate the existence of such phenomena empirically, analyze the root cause for it theoretically, and propose a simple yet effective normalization method to tackle the problem. Several experiments demonstrate the effectiveness of this method.

In general I think the paper is descent and elegant. It is motivated from real-world pain-point, gives a rigorous study towards the root cause, and the proposed method is very effective. To the best of my knowledge there is no prior work looking deep into this area and this paper does bring new insights to the community. As a result I would vote for its acceptance.

One issue is that I find the backbone methods in experiments are somehow out-of-date. For example, AutoRec (2015) and CF-NADE (2016). I admit that I’m not an expert in the field of recommendation but still think that more recent, and powerful baseline algorithms should be applied on to further demonstrate the true effectiveness of Sparsity Normalization.


**Experience Assessment:**

I have published one or two papers in this area.

**Review Assessment: Checking Correctness Of Derivations And Theory:**

I assessed the sensibility of the derivations and theory.

**Review Assessment: Checking Correctness Of Experiments:**

I assessed the sensibility of the experiments.

**Review Assessment: Thoroughness In Paper Reading:**

I made a quick assessment of this paper.

---

> ### Author Response · Authors · 2019-11-13
> **Response to Reviewer #2**
>
> We thank the reviewer for thoughtful and constructive feedback.
>
> [Powerful backbone architecture on collaborative filtering datasets]
> - We use AutoRec (Sedhain et al., 2015) and its variant CF-NADE (Zheng et al., 2016) without any intention simply because many modern nn-based models are in fact variants of AutoRec. But, following the reviewers’ valuable suggestion, we consider CF-UIcA [3], one of the current state-of-the-arts, as a new backbone on collaborative filtering datasets, and consistently achieve even stronger performances (in terms of RMSE):
>     - Movielens 100K: 0.8945 ± 0.0024 (w/o SN) vs. 0.8793 ± 0.0017 (w/ SN)
>     - Movielens 1M: 0.8223 ± 0.0016 (w/o SN) vs. 0.8178 ± 0.0007 (w/ SN)
>
> Note that we do not test for Movielens 10M because the authors of CF-UIcA did not provide the results for it due to the complexity of the model.
>
> [3] Du, C., Li, C., Zheng, Y., Zhu, J., & Zhang, B. (2018, April). Collaborative filtering with user-item co-autoregressive models. In Thirty-Second AAAI Conference on Artificial Intelligence.

---

### Public Comment · ~Jaeyoon_Yoo1 · 2019-10-27
**need to compare with other imputation method**

Hi, it's interesting paper for handling missing data.

But, it needs more thorough comparison

As far as I understand correctly, w/o SN is zero imputation and w/ SN is your variant of zero imputation.

Then, there should be comparison between yours and  other imputation methods.

One example is Processing of missing data by neural networks, 2018 NIPS.

"dropout" in the paper seems to correspond to SN, but it showed almost always worse performance even than k-nn imputation.

Although I mentioned only one, there are many other imputations to handle missing data, so they also should be covered.

Thanks

---

> ### Author Response · Authors · 2019-11-13
> **Response for Jaeyoon Yoo**
>
> Thank you for your comments. Please refer to the answer to R1 or R3.
>
> There is a minor note about the dropout method in the GMMC [2] paper you mentioned. The dropout method uses a single drop (missing) probability uniformly across all instances of the dataset. On the other hand, our algorithm normalizes each data instance with its own missing rate. The global drop probability does not solve the variable sparsity problem, so it shows poor performance as seen in Appendix H. For example, on Movielens 100K dataset (AutoRec with item vector encoding), the RMSE of dropout and SN is as follows: 0.9268 ± 0.0261 (dropout) vs 0.8809 ± 0.0011 (zero imputation w/ SN). For other various datasets and models, SN also shows similar or better performance compared to the dropout.
>
> [2] Marek ́Smieja, Łukasz Struski, Jacek Tabor, Bartosz Zieli ́nski, and Przemysław Spurek. Processing of missing data by neural networks. In Advances in Neural Information Processing Systems, pp.2719–2729, 2018.

---

### Decision · Program_Chairs · 2019-12-19

**Decision:**

Accept (Poster)

**Comment:**

This paper investigates the problem of using zero imputation when input features are missing. The authors study this problem, propose a solution, and evaluate on several benchmark datasets. The reviewers were generally positive about the paper, but had some questions and concerns about the experimental results. The authors addressed these concerns in the rebuttal. The reviewers are generally satisfied and believe that the paper should be accepted.